# A widespread family of heat-resistant obscure (Hero) proteins protect against protein instability and aggregation

Kotaro Tsuboyama[1,2]*, Tatsuya Osaki[3,4], Eriko Matsuura-Suzuki[2,5], Hiroko Kozuka-Hata[6], Yuki Okada[7], Masaaki Oyama[1,6], Yoshiho Ikeuchi[3,4], Shintaro Iwasaki[1,5], Yukihide Tomari[1,2]*

1 Department of Computational Biology and Medical Sciences, Graduate School of Frontier Sciences, The University of Tokyo, Bunkyo-ku, Tokyo, Japan, 2 Laboratory of RNA Function, Institute for Quantitative Biosciences, The University of Tokyo, Bunkyo-ku, Tokyo, Japan, 3 Biomolecular and Cellular Engineering laboratory, Institute of Industrial Science, The University of Tokyo, Meguro-ku, Tokyo, Japan, 4 Department of Chemistry and Biotechnology, School of Engineering, The University of Tokyo, Bunkyo-ku, Tokyo, Japan, 5 RNA Systems Biochemistry Laboratory, RIKEN Cluster for Pioneering Research, Wako, Saitama, Japan, 6 Medical Proteomics Laboratory, The Institute of Medical Science, The University of Tokyo, Minato-ku, Tokyo, Japan, 7 Laboratory of Pathology and Development, Institute for Quantitative Biosciences, The University of Tokyo, Bunkyo-ku, Tokyo, Japan

* tsuboyama-tky@umin.ac.jp (KT); tomari@iam.u-tokyo.ac.jp (YT)

**Data Availability Statement:** All relevant data are within the paper and its Supporting Information files.

## Abstract

Proteins are typically denatured and aggregated by heating at near-boiling temperature. Exceptions to this principle include highly disordered and heat-resistant proteins found in extremophiles, which help these organisms tolerate extreme conditions such as drying, freezing, and high salinity. In contrast, the functions of heat-soluble proteins in non-extremophilic organisms including humans remain largely unexplored. Here, we report that heat-resistant obscure (Hero) proteins, which remain soluble after boiling at 95°C, are widespread in *Drosophila* and humans. Hero proteins are hydrophilic and highly charged, and function to stabilize various "client" proteins, protecting them from denaturation even under stress conditions such as heat shock, desiccation, and exposure to organic solvents. Hero proteins can also block several different types of pathological protein aggregations in cells and in *Drosophila* strains that model neurodegenerative diseases. Moreover, Hero proteins can extend life span of *Drosophila*. Our study reveals that organisms naturally use Hero proteins as molecular shields to stabilize protein functions, highlighting their biotechnological and therapeutic potential.

## Introduction

Proteins are polymers composed of 20 different amino acids, whose side chains have various properties such as aliphatic, aromatic, acidic, basic, and sulfur-containing. This diversity allows proteins to fold into three-dimensional structures, which determine their activity and function. Although proteins are generally stable at the physiological temperature or even at

**Funding:** This work was supported in part by JSPS KAKENHI Grant Number JP16J02134 and 19J30003 to KT (https://www.jsps.go.jp), and MEXT KAKENHI Grant Number JP26113007 (http://www.mext.go.jp) and JSPS KAKENHI Grant Number 18H05271 to YT. The funders had no role in study design, data collection and analysis, decision to publish, or preparation of the manuscript.

**Competing interests:** I have read the journal's policy and the authors of this manuscript have the following competing interests: YT, KT, SI, EM-S, HK-H, and MO have a patent application related to this work. The other authors declare no competing interests.

**Abbreviations:** AGO, Argonaute; ALS, amyotrophic lateral sclerosis; BDNF, brain-derived neurotrophic factor; BSA, bovine serum albumin; CRISPR, clustered regularly interspaced short palindromic repeat; DIC, differential interference contrast; DIOPT, DRSC Integrative Ortholog Prediction Tool; EIF4A1, eukaryotic initiation factor 4A-I; FDR, false discovery rate; FTD, frontotemporal dementia; GFP, green fluorescent protein; GMR, glass multimer reporter; GOT1, aspartate aminotransferase; gRNA, guide RNA; GST, glutathione S-transferase; Hero, heat-resistant obscure; HRP, horseradish peroxidase; Hsp, heat shock protein; IDP, intrinsically disordered protein; IDR, intrinsically disordered region; iPS, induced-pluripotent stem; KO, knockout; KR, lysine and arginine; LC-MS/MS, liquid chromatography-mass spectrometry; LDH, lactate dehydrogenase; LEA, late embryogenic abundant; LLPS, liquid-liquid phase separation; MJDtr-Q78, Machado-Joseph disease truncated-polyglutamine 78; PAS, proline, alanine, and serine; PEG, polyethylene glycol; PGK1, phosphoglycerate kinase 1; pI, isoelectric point; PSM, peptide spectrum match; RA, retinoic acid; RHP, random heteropolymer; RISC, RNA-induced silencing complex; RPKM, reads per kilobase of exon per million mapped sequence reads; TAR, *trans*-activation response element; TDP, tardigrade disordered protein; TDP-43ΔNLS, TDP-43 lacking the nuclear localization signal; TEV, tobacco etch virus; Ub, ubiquitin; YFP, yellow fluorescent protein.

approximately 50–60°C [1], heating at near-boiling temperature destroys structures and induces denaturation and aggregation for most proteins. However, there are some exceptions. For example, tardigrade disordered proteins (TDPs), found in the supernatant of tardigrade lysate after boiling [2], are required for tardigrades to survive desiccation [3]. Another example is late embryogenic abundant (LEA) proteins in plants, whose expression is associated with their tolerance to dehydration, freezing, or high salinity [4,5]. LEA proteins are also found in extremophiles including the radiation-resistant bacterium *Deinococcus radiodurans* and desiccation-tolerant animals such as *Artemia*, *Caenorhabditis elegans*, and rotifers [6,7]. TDPs and most LEA proteins are extremely hydrophilic and heat soluble, and can be classified into intrinsically disordered proteins (IDPs). Currently, these heat-resistant proteins are viewed as special cases required by organisms living in extreme conditions to protect their functional proteins. Although mammals have been reported to produce some highly heat-soluble proteins [8,9], their functions remain largely unexplored.

Here, we show that a class of heat-soluble proteins, which we call "Hero" (heat-resistant obscure) proteins, are widespread in non-extremophilic animals such as *Drosophila* and humans. Hero proteins act as molecular shields to protect various "client" proteins from denaturation even under stress conditions, such as heat shock, desiccation, and exposure to organic solvents. Moreover, Hero proteins can suppress several forms of pathogenic protein aggregates in cells and in in vivo models for neurodegenerative diseases, and extend life span of *Drosophila*. Our findings not only have significant implications for our fundamental understanding of protein stability and functions but also highlight potential biotechnological and therapeutic applications of Hero proteins.

## Results

### Boiled supernatants of crude cell lysates improve Ago2 molecular behavior

Many proteins are naturally unstable even under physiological conditions, probably because they need to prioritize function over structural durability. For example, Argonaute (AGO) proteins, the core of RNA-induced silencing complex (RISC), must load a small RNA duplex, eject the passenger strand from the two strands of the duplex, recognize target RNAs complementary to the remaining guide strand, and endonucleolytically cleave RNA targets or recruit downstream silencing factors [10]. AGO proteins are highly unstable; in particular, empty AGO is structurally flexible at the single-molecule level [11] and susceptible to proteolytic digestion in vitro [12]. During the course of our studies on AGO, we observed a strange phenomenon: when immunopurifying *Drosophila* Ago2 fused with a tobacco etch virus (TEV)-cleavable FLAG-tag using anti-FLAG magnetic beads, we were unable to elute the protein from the beads, even after the FLAG-tag was removed using TEV protease (Fig 1A and 1B). We concluded that purified, free Ago2 protein is unstable and tends to stick nonspecifically to the beads. Strikingly, addition of crude *Drosophila* S2 cell lysate during the TEV protease treatment promoted efficient elution of Ago2 (Fig 1B). To test whether a protein(s) was responsible for this effect, we boiled the crude S2 cell lysate at 95°C for 15 minutes and removed the precipitated proteins by centrifugation. Unexpectedly, the supernatant of the heat-denatured lysate, which contained only approximately 4.3% of the total protein concentration in the original crude lysate, was as competent in eluting Ago2 as the original lysate. Digestion of proteins in the boiled supernatant with proteinase K, followed by its inactivation by a second round of heating, halved its activity in Ago2 elution (Fig 1B). To explore what materials other than proteins contribute to Ago2 elution, we utilized benzonase to deplete DNAs and RNAs. Treatment of the boiled lysate by both proteinase K and benzonase abolished the elution activity, while the benzonase treatment alone only modestly decreased the activity (Fig 1B). Elution of

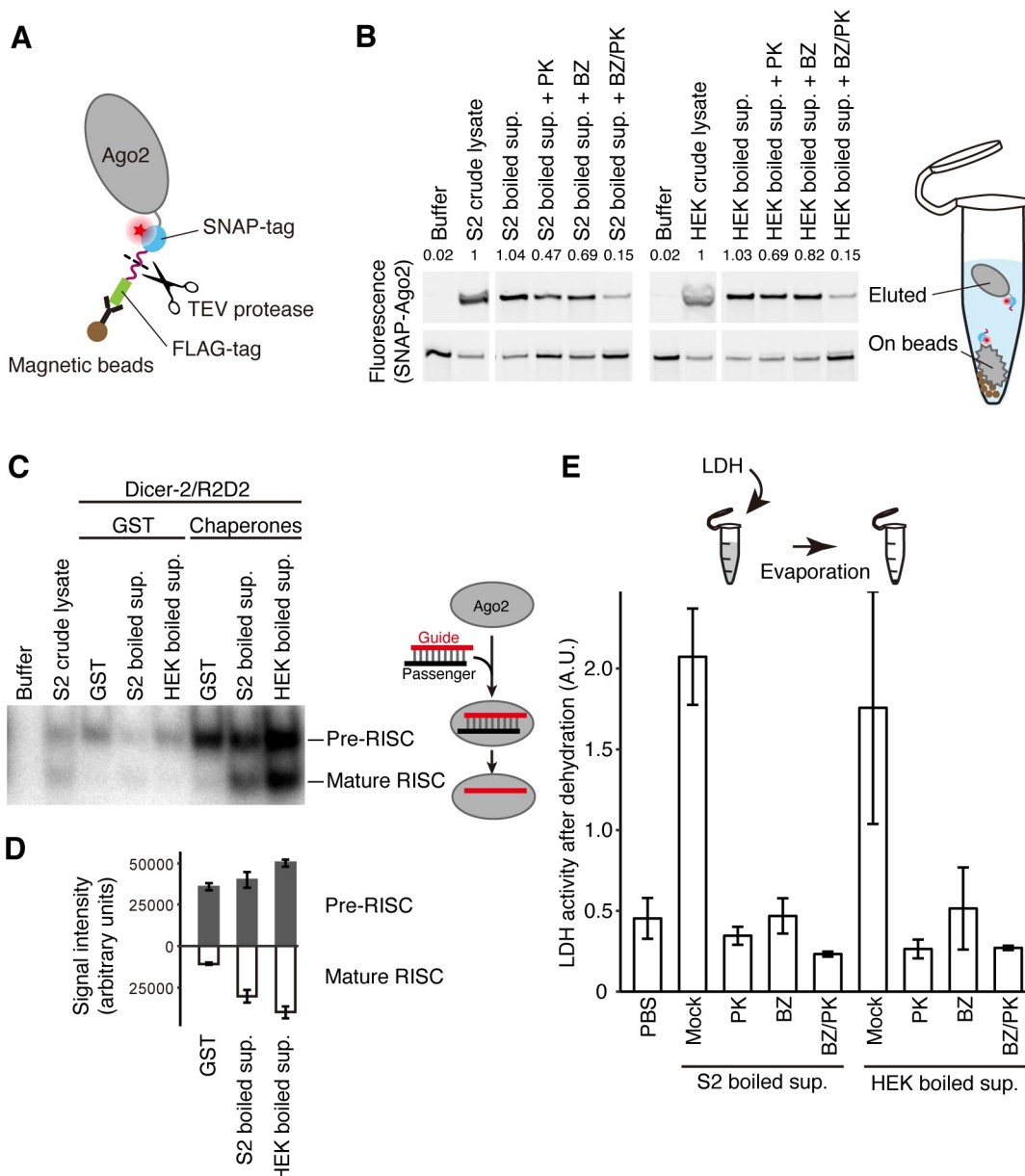

**Fig 1. Activity of heat-soluble proteins and nucleonic acids on Ago2 molecular behavior and LDH desiccation.** (A) Schematic representation of FLAG-TEV-SNAP-Ago2 immunopurified on magnetic beads. (B) FLAG-TEV-SNAP-Ago2 was immunopurified onto magnetics beads via anti-FLAG antibody, and then the FLAG-tag was cleaved off by TEV protease in buffer, crude lysates, or their boiled supernatants from fly S2 or human HEK293T cells. PK indicates that the boiled supernatant had been mostly deproteinized by proteinase K in advance. BZ indicates that DNA and RNA in boiled supernatant had been degraded by benzonase. BZ/PK indicates that DNA and RNA had been degraded by benzonase, followed by the deproteinization by proteinase K. Eluted Ago2 (top) and Ago2 still remaining on the beads (bottom) were visualized by a red fluorescent dye covalently attached to the SNAP tag. The values on the top indicate relative amounts of eluted Ago2 normalized to that with S2 crude cell lysate. Proteins remaining in the boiled supernatants have an activity to promote Ago2 elution. (C) Small RNA pull-down assay for pre-Ago2-RISC and mature Ago2-RISC assembled in the reconstitution system, containing a $^{32}$P-radiolabeled small RNA duplex, Ago2, Dicer-2/R2D2, and the Hsp70/Hsp90 chaperone machinery. Supplementation of the boiled supernatants from S2 or HEK293T cells promoted the formation of both pre- and mature Ago2-RISC. (D) Quantification of (C). Data represent means ± SD from 3 independent experiments. (E) Heat-soluble proteins in the boiled supernatants protect the LDH activity from desiccation. LDH mixed with the indicated boiled supernatants or buffer was dried up overnight, and the remaining LDH activities were measured. Data represent means ± SD from 3 independent experiments. The numerical data pertaining to this figure can be found in S1 Data file. Ago, Argonaute; GST, glutathione S-transferase; Hsp, heat shock protein; LDH, lactate dehydrogenase; RISC, RNA-induced silencing complex; TEV, tobacco etch virus.

*Drosophila* Ago2 was similarly promoted by crude lysate from human HEK293T cells and by its boiled supernatant (approximately 6.0% of the original protein concentration). Again, the efficiency was nearly halved by the proteinase K treatment, modestly decreased by benzonase, and abrogated by proteinase K and benzonase together (Fig 1B). These results suggest that some heat-soluble proteins as well as nucleic acids contained in the boiled S2 or HEK293T cell supernatants promote elution of free Ago2 from the beads.

We asked whether the boiled supernatants were capable of influencing any of the functional features of Ago2. Assembly of *Drosophila* Ago2-RISC requires not only a small RNA duplex and Ago2 itself but also the Dicer-2/R2D2 heterodimer and the heat shock protein (Hsp)70/90 chaperone machinery [13]. All these factors are present in crude S2 cell lysate, and incubation of FLAG-tagged Ago2 immobilized on magnetic beads with a $^{32}$P-radiolabeled small RNA duplex and S2 cell lysate efficiently assembled pre-RISC, containing both strands of the duplex, and mature RISC, containing only the single-stranded guide (Fig 1C). Compared to the crude S2 lysate, reconstitution of Ago2-RISC assembly using high concentrations of purified Hsp70/90 chaperone machinery and Dicer-2/R2D2 efficiently formed pre-Ago2-RISC, but its conversion into mature Ago2-RISC was inefficient. Adding the supernatant of heat-denatured S2 or HEK293T cell lysates strongly promoted the formation of both pre- and mature Ago2-RISC, although without the chaperones, the supernatants themselves showed no reconstitution activity (Fig 1C and 1D). Thus, the boiled supernatants from either fly S2 or human HEK293T cells can promote the assembly of Ago2-RISC by Dicer-2/R2D2 and the chaperone machinery. Together, our data suggest that factors in the boiled supernatants improve the molecular behavior of Ago2, either in its free form (Fig 1A and 1B) or upon RISC assembly (Fig 1C and 1D), and that these factors act beyond the species boundary.

### *Drosophila* and human heat-soluble proteins protect lactate dehydrogenase activity from desiccation

We next asked whether heat-soluble proteins in the boiled supernatants could act generally to enhance or protect the activity of other proteins, especially in stress conditions. We first examined the effect of boiled supernatants on protein dehydration. When purified lactate dehydrogenase (LDH) from rabbit muscle was dried up at room temperature overnight, addition of the boiled supernatants from S2 or HEK293T cells efficiently preserved the LDH activity compared to the PBS buffer only condition (Fig 1E). This effect was eliminated by proteinase K and/or benzonase treatment of the supernatants (Fig 1E), indicating that heat-soluble *Drosophila* and human proteins as well as nucleic acids can protect the activity of LDH from desiccation. Because it is already known that nucleic acids participate in maintaining protein homeostasis [14,15], we decided to focus on heat-soluble proteins in the following analyses.

### Hydrophilic and highly charged proteins are enriched in the boiled supernatants

Heat generally denatures protein structures by exposing hydrophobic regions. Thus, proteins remaining soluble in the boiled supernatant are likely to be hydrophilic and intrinsically disordered. To test this idea, we subjected the crude lysates and their boiled supernatants of fly S2 and human HEK293T cells to liquid chromatography-mass spectrometry (LC-MS/MS) analysis. In total, we identified 910 (S2) and 980 (HEK293T) proteins with high confidence (false discovery rate [FDR] < 0.01). We then divided the identified proteins into five groups according to the degrees of depletion or enrichment by boiling (Group I [most depleted] to Group V [most enriched]), in such a way that each group contained virtually the same number of proteins (Fig 2A and 2B, and S1 and S2 Tables; also see Materials and methods). To examine the

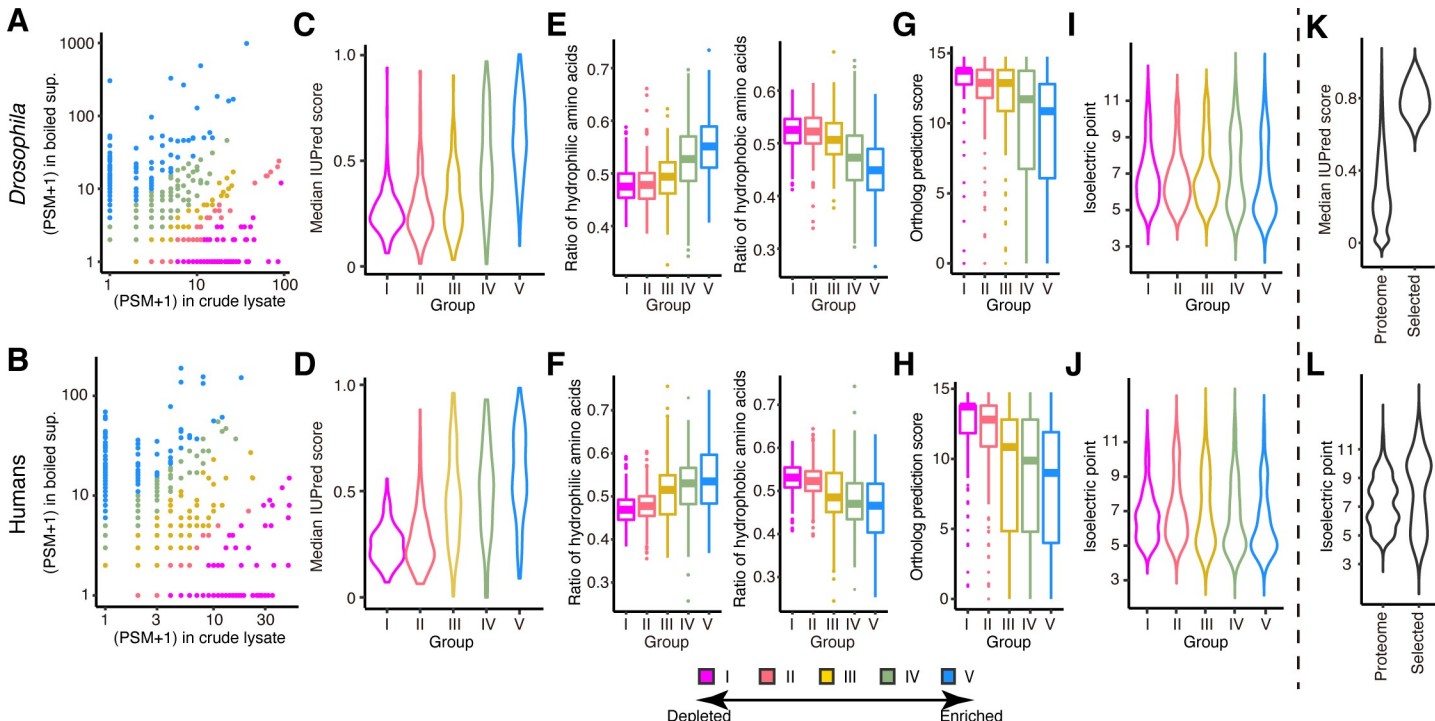

**Fig 2. Hydrophilic and highly charged proteins are enriched in the boiled supernatants.** (A–B) LC-MS/MS analysis of lysates from *Drosophila* S2 (A) and human HEK293T (B) cells before and after boiling. Scatterplots indicate the PSM values (plus 1, to avoid zeros), which correlate with the protein amount, in the original crude lysates on the x-axes and those in the boiled supernatants on the y-axes. The identified proteins were divided into five groups according to the degrees of depletion or enrichment by boiling (Group I [most depleted] to Group V [most enriched]) and used for further analysis in C–J. (C–D) Violin plots of the median IUPred score of the proteins classified into the five groups in *Drosophila* (C) and humans (D). The proteins enriched by boiling are predicted to be much more disordered. (E–F) Boxplots of the proportion of hydrophilic or hydrophobic amino acids of the proteins classified into the five groups in *Drosophila* (E) and humans (F). The proteins enriched by boiling are more hydrophilic. Boxplot center and box edges indicate median and 25th or 75th percentiles, respectively, while whiskers indicate the median ± 1.5 × IQR or the most extreme observations within these limits, and outliers are plotted individually. (G–H) Boxplots of the DIOPT score of the proteins classified into the five groups in *Drosophila* (G) and humans (H). The proteins enriched by boiling are predicted to be evolutionarily less conserved. Boxplot center and box edges indicate median and 25th or 75th percentiles, respectively, while whiskers indicate the median ± 1.5 × IQR or the most extreme observations within these limits, and outliers are plotted individually. (I–J) Violin plots of the pI values of the proteins classified into the five groups in *Drosophila* (I) and humans (J). The proteins enriched by boiling show the pI values concentrated in the acidic (approximately 5) or basic (approximately 10) range but avoiding the neutral (between 7 and 8) range. (K) Violin plots of the median IUPred score of the entire human proteome and that of the proteins selected by their high IUPred scores and expression levels. (L) Violin plots of the pI values of the entire human proteome and that of the highly disordered proteins selected by their high IUPred scores and expression levels. The selected proteins also show the pI values concentrated in the acidic or basic range but avoiding the neutral range. The PSM data pertaining to this figure can be found in S1 and S2 Tables files. DIOPT, DRSC Integrative Ortholog Prediction Tool; LC-MS/MS, liquid chromatography-mass spectrometry; pI, isoelectric point; PSM, peptide spectrum match.

overall characteristics of heat-soluble proteins, we utilized a prediction tool for intrinsically disordered regions (IDRs) called IUPred, whose score above 0.5 indicates that the amino acid residue at a given position is likely to be a part of a disordered region [16]. For simplicity, we calculated the median IUPred score of all the residues for each protein. As expected, the proteins enriched by boiling showed much higher distributions of the median IUPred score than the depleted ones in both *Drosophila* or humans (Fig 2C and 2D). Moreover, hydrophilic amino acids were overrepresented, while hydrophobic amino acids were underrepresented in the boil-enriched proteins (Fig 2E and 2F). Thus, structurally disordered, hydrophilic proteins were indeed enriched in the boiled supernatant.

Motifs previously associated with LEA proteins were not found among the proteins identified in the boiled supernatants. In fact, no common or conserved motifs were detected among their primary sequences. Moreover, the DRSC Integrative Ortholog Prediction Tool (DIOPT) score, which integrates 15 different ortholog prediction tools for the comparison between *Drosophila* and humans [17], was gradually decreased as the enrichment by boiling was increased,

indicating that heat-soluble proteins are evolutionarily less conserved in their primary amino acid sequences than regular heat-vulnerable proteins (Fig 2G and 2H). These data suggest that heat-soluble proteins as a whole cannot be defined by conventional, sequence-, and homology-based classifications. Instead, we found that the isoelectric point (pI) values of boil-enriched proteins have a bimodal distribution, concentrating on the acidic (approximately 5) or basic (approximately 10) region but avoiding the neutral (between 7 and 8) region (Fig 2I and 2J). Thus, heat-resistant proteins tend to be either highly negatively charged or highly positively charged under the physiological pH.

Because the LC-MS/MS analysis has an intrinsic bias that depends on the size range of protease-digested peptide fragments and can only detect proteins expressed in HEK293T cells, we also utilized the IUPred score itself, independently of the MS-identified list, to select proteins from the entire human proteome for further functional analyses. In the human protein atlas [18], we found approximately 450 proteins (Fig 2K) that have both high IUPred scores across their entire sequences (median IUPred > 0.6, excluding locally structured proteins with >25% of residues having an IUPred score of <0.4) and high expression levels (median reads per kilobase of exon per million mapped sequence reads [RPKM] > 25), which also showed a bimodal pI distribution (Fig 2K and 2L). Of these, we chose six representative proteins (C9orf16, C11orf58, BEX3, SERBP1, SERF2, and C19orf53; S1A and S1B Fig) whose mRNA levels are as high as that of the most abundant form of Hsp70 (HSPA8) in HEK293T cells, four of which (C9orf16, C11orf58, SERBP1, and SERF2) were detected by the LC-MS/MS analysis of the boiled supernatant. The pI values for the six proteins were 4.13 (C9orf16), 4.57 (C11orf58), 5.31 (BEX3), 8.66 (SERBP1), 10.44 (SERF2), and 11.55 (C19orf53), respectively.

We next assayed the heat solubility of these six candidate proteins, together with green fluorescent protein (GFP) and glutathione S-transferase (GST) as controls (Fig 3A). We expressed them as FLAG-tagged proteins in HEK293T cells, boiled the cell lysate at 95˚C for 15 minutes, then examined the supernatant by anti-FLAG western blotting. All of six proteins were readily detectable in the soluble supernatant after boiling, whereas GFP and GST were essentially eliminated from the boiled supernatant (Fig 3A). Based on their unusual heat resistance and unstructured nature, we call this protein class heat-resistant obscure (Hero) proteins (*hero-hero* in Japanese means flimsy, loose, or flexible) and add the molecular weight to distinguish each protein, following the naming convention for heat shock proteins (e.g., Hsp70). Hereafter, we call C9orf16, C11orf58, BEX3, SERBP1, SERF2, and C19orf53 as Hero9, -20, -13, -45, -7, and -11, respectively, for convenience.

## Hero proteins protect the activity of various proteins under stress conditions

To evaluate the function of each of the six representative Hero proteins, we produced recombinant human Hero proteins in *Escherichia coli*. We then mixed 5 μg/mL of LDH with 4 μg/mL of each Hero protein and subjected the mixture to overnight dehydration. Like the boiled supernatants (Fig 1E), all six Hero proteins protected the LDH activity at approximately 50%, which was much stronger compared to bovine serum albumin (BSA), GST, or the conventional protein stabilizers arginine or trehalose (Fig 3B). The LDH-protecting activity under desiccation is similar to what was previously observed for TDPs [3], which share no common sequence motifs with human Hero proteins.

We next explored the protective effect of Hero proteins in organic solvents, a drastically harsh and foreign condition for proteins. When GFP was exposed to chloroform, its fluorescent intensity was reduced to approximately 10% of the native state (Fig 3C). In contrast, the addition of four out of six Hero proteins maintained >60% of the GFP activity (Fig 3C). Strikingly,

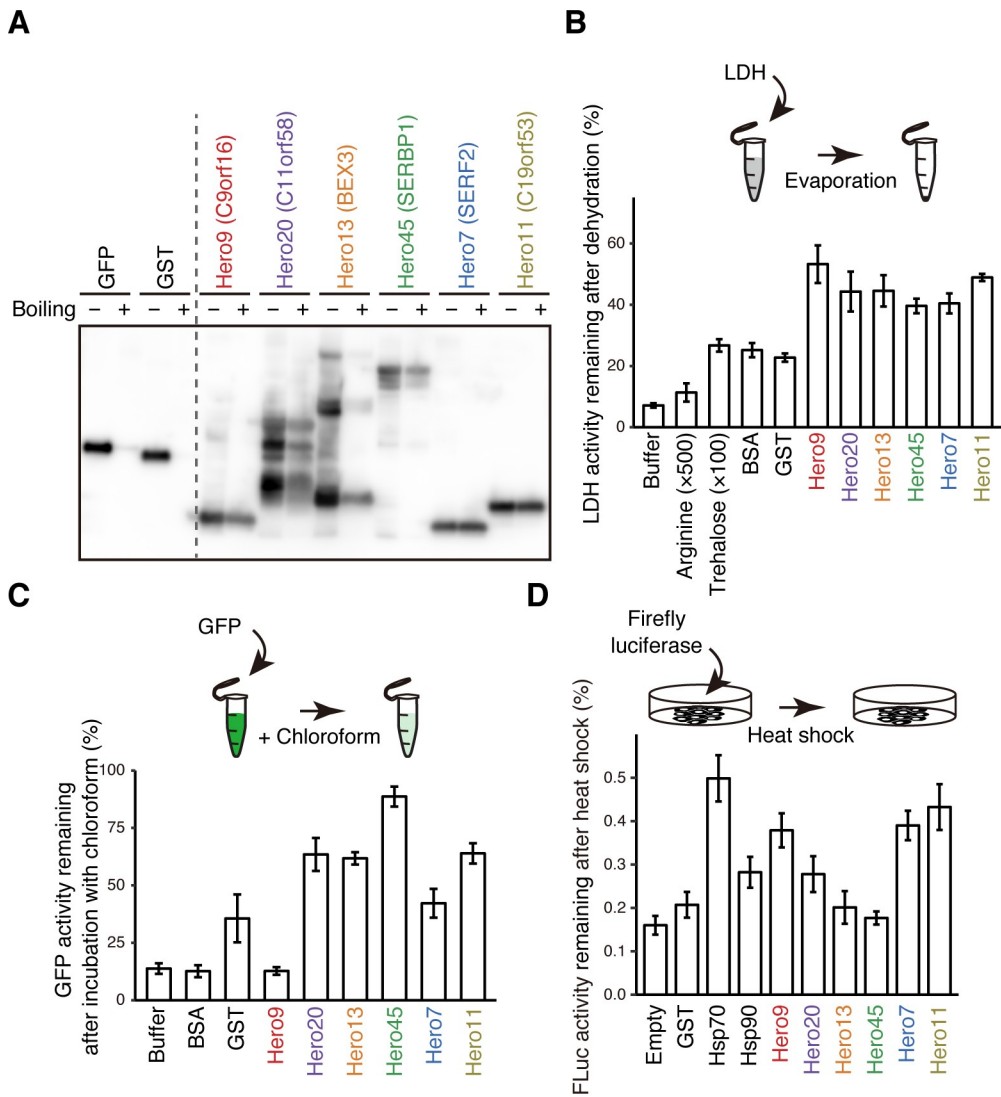

**Fig 3. Hero proteins protect the activity of various proteins in stress conditions.** (A) Heat solubility of the six Hero proteins selected. Each Hero protein, GFP, or GST was fused with a FLAG-tag and expressed in HEK293T cells. The cell lysates were prepared with (+) or without (−) boiling, and then the proteins in the lysates or their supernatants were detected by anti-FLAG western blotting. The result was confirmed by two independent experiments. (B) Hero proteins protect the LDH activity from desiccation. LDH mixed with the indicated Hero protein or controls was dried up overnight and the remaining LDH activities were measured. Note that much higher concentrations of arginine (500-fold) and trehalose (100-fold) were used, compared to those of the proteins. Data were normalized to the activity of LDH incubated on ice overnight and represent means ± SD from 3 independent experiments. (C) Hero proteins protect GFP in organic solvent. GFP mixed with the indicated Hero protein or controls was exposure to chloroform and the remaining GFP fluorescence was measured. Data were normalized to the activity of GFP without the exposure to chloroform and represent means ± SD from 3 independent experiments. (D) Hero proteins protect the luciferase activity from heat shock. Firefly luciferase was transfected into HEK293T cells, together with each Hero protein or chaperone factor, and then the cells were exposed to heat shock. Remaining luciferase activities after heat shock were measured. Data were normalized to the activity of luciferase in cells without the exposure to heat shock, and represent means ± SD from 5 independent experiments. The numerical data pertaining to this figure can be found in S1 Data file. BSA, bovine serum albumin; GFP, green fluorescent protein; GST, glutathione S-transferase; Hero, heat-resistant obscure; Hsp, heat shock protein; LDH, lactate dehydrogenase.

in the presence of Hero45, GFP fluorescence remained undiminished even in chloroform (Fig 3C). This is reminiscent of the ability of methacrylate-based random heteropolymers (RHPs) to

preserve horseradish peroxidase (HRP) activity in toluene [19]. Thus, naturally occurring Hero proteins act as molecular shields to stabilize proteins, even against harsh conditions.

To investigate the effect of Hero proteins in cells, we expressed firefly luciferase together with each of the six Hero proteins, Hsp70, or Hsp90 in HEK293T cells, then subjected the cells to heat shock at 45°C for 8 minutes. Heat shock reduced the luciferase activity to between 10% and 20% of the original level when co-transfected with an empty or GST-expressing vector. Overexpression of the conventional chaperones, Hsp70 or Hsp90, maintained between 30% and 50% of luciferase activity (Fig 3D). Remarkably, Hero proteins protected luciferase from the heat shock as well as the chaperones, with Hero9, -7, and -11 preserving approximately 40% of luciferase activity (Fig 3D). We concluded that Hero proteins have the ability to stabilize various proteins both in vitro and in cells.

## Hero proteins prevent pathogenic protein aggregations in cells

Protein instability is often linked to diseases, especially neurodegenerative disorders. For example, aggregation of trans-activation response element (TAR) DNA-binding protein of 43 kDa (TDP-43) is observed in virtually all cases of amyotrophic lateral sclerosis (ALS) and in about half the cases of frontotemporal dementia (FTD) [20]. Given the strong activity of Hero proteins to stabilize proteins (Figs 1–3), we tested if Hero proteins could prevent pathogenic aggregation of TDP-43. We constructed TDP-43 lacking the nuclear localization signal (TDP-43ΔNLS), which is more prone to aggregation than the wild type because of its forced cytoplasmic localization. We expressed TDP-43ΔNLS as a GFP fusion in HEK293T cells, together with each of the six human Hero proteins or GST as a control. Aggregates of GFP-TDP-43ΔNLS and their suppression by co-expression of Hero proteins were apparent at the microscopic level (Fig 4A). To quantitatively analyze the degree of aggregation, the cells were lysed by sonication, and aggregates in the extracts were measured by a filter trap assay using a 0.2 μm cellulose acetate membrane in the presence of 1% SDS. We found that Hero45, Hero7, and Hero11 strongly suppressed TDP-43ΔNLS aggregation, compared to typical structured proteins such as GST, firefly luciferase (Luc), phosphoglycerate kinase 1 (PGK1), eukaryotic initiation factor 4A-I (EIF4A1), aspartate aminotransferase (GOT1), and ubiquitin (Ub) (Fig 4B and S2 Fig). In addition to TDP-43, we also analyzed two additional model proteins that use different mechanisms to form pathological aggregates. HTTQ103 consists of a stretch of 103 polyglutamine residues deriving from the abnormal CAG expansion found in Huntington disease–causing huntingtin mutant [21]. GA50 (50 Glycine-Alanine repeats) is derived from the abnormal GGGGCC repeats found in the ALS-causing C9orf72 intron [22]. Both at the microscopic level and by the filter trap assay, aggregates of HTTQ103 and GA50 were also markedly reduced by Hero9 (HTTQ103), Hero45 (GA50), and several other Hero proteins (Fig 4A and 4B and S2 Fig). Of note, prevention of pathogenic polyglutamine and polyalanine aggregations in the T-REx293 cell line was previously shown for LEA proteins derived from an anhydrobiotic nematode or soya bean [23,24]. Thus, anti-aggregation may be a common feature for heat-soluble proteins.

Interestingly, among the six Hero proteins tested, there was no "super-Hero" protein able to function for all proteins. Instead, distinct subsets of human Hero proteins were more effective in preventing different types of aggregations (Fig 4B). For example, Hero11 promoted rather than prevented aggregation of GA50 compared to GST, while strongly suppressing that of TDP-43ΔNLS and HTTQ103 (Fig 4B). Similarly, Hero7 was previously shown to promote aggregation of HA-tagged HTTQ74 in SK-N-SH cells [25], whereas we found that it strongly suppressed aggregation of GFP-tagged TDP-43ΔNLS in HEK293T cells (Fig 4B). We suggest that different Hero proteins have different preferences for their protein "clients," which may

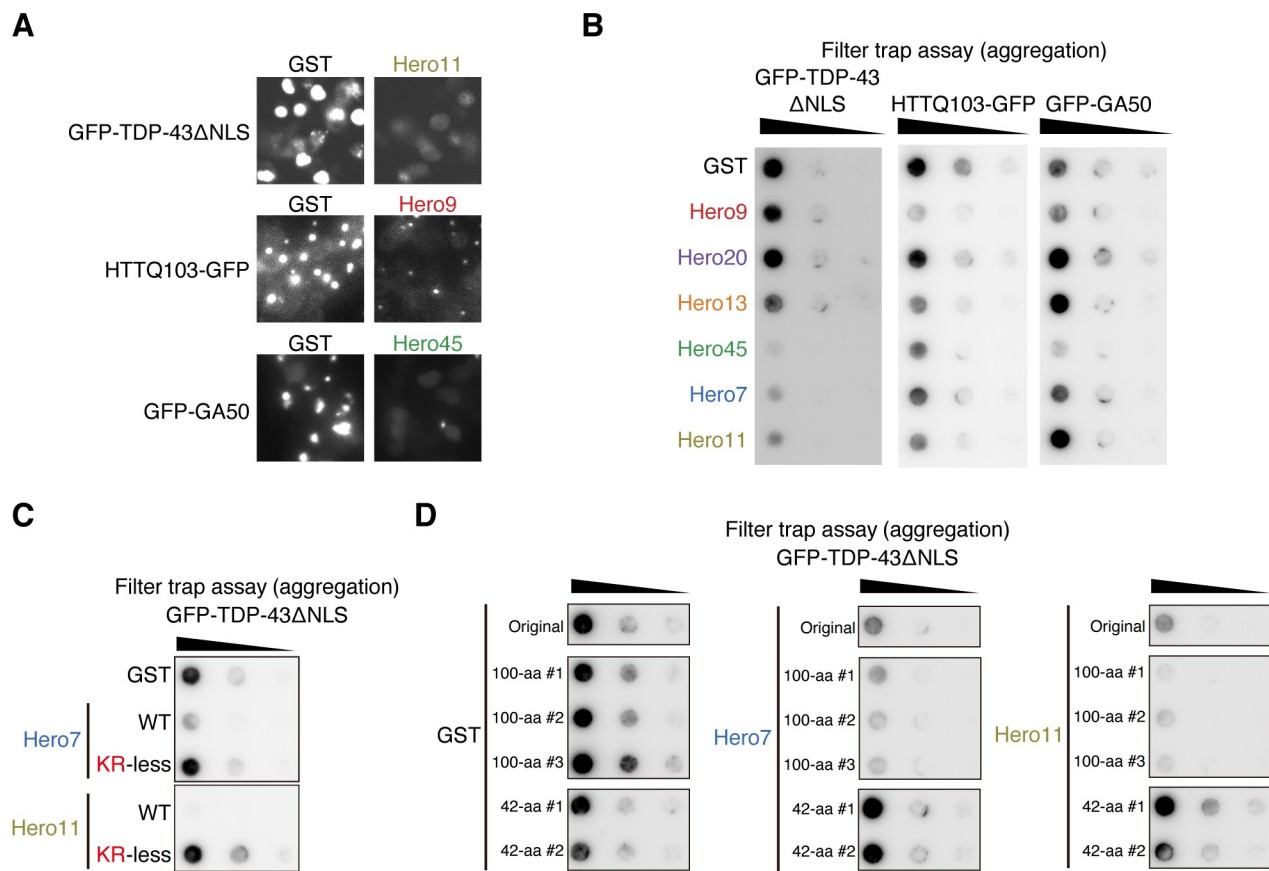

**Fig 4. Hero proteins prevent pathogenic protein aggregation in cells.** (A) Representative microscopic images of GFP signals in the cells transfected with GFP-tagged TDP-43ΔNLS, HTTQ103, or GA50, together with the GST control or each Hero protein. Note that signals from aggregations are saturated. (B) Filter trap assay of aggregation-prone proteins expressed in cells. TDP-43ΔNLS, HTTQ103, or GA50 was expressed in a GFP-fusion form in HEK293T cells, together with each Hero protein or the GST control. The original or 5-fold diluted samples were loaded on a cellulose acetate membrane in the presence of 1% SDS and the trapped aggregates were probed with anti-GFP antibody. (C) Filter trap assay of aggregation-prone proteins expressed in cells. GFP-TDP-43ΔNLS was expressed in HEK293T cells, together with each Hero protein, its KR-less mutant, in which positively charged amino acids (Lys and Arg) were substituted with neutral Gly, or the GST negative control. Five-fold serial dilutions of the cell lysates were subjected to the filter trap assay and probed with the anti-GFP antibody. Experiments were simultaneously performed, and the presented data were spliced from the same membrane. (D) Filter trap assay of aggregation-prone proteins expressed in cells. GFP-TDP-43ΔNLS was expressed in HEK293T cells, together with the GST control, each Hero protein, or their shuffled amino acid sequences with 42– or 100–amino acid length. Five-fold serial dilutions of the cell lysates were subjected to the filter trap assay and probed with anti-GFP antibody. Experiments were simultaneously performed, and the presented data were spliced from the same membrane. aa, amino acid; GFP, green fluorescent protein; GST, glutathione S-transferase; Hero, heat-resistant obscure; KR, lysine and arginine; TDP-43ΔNLS, TDP-43 lacking the nuclear localization signal; WT, wild-type.

vary depending on the cellular environments. Nevertheless, it is remarkable that, for a variety of assays with different client proteins and experimental settings (Figs 3 and 4), there were always multiple Hero proteins (despite looking at only six out of hundreds of putative Hero proteins in humans) that showed strong protein-stabilization and anti-aggregation effects.

## Extreme charges of Hero proteins are necessary and sufficient to suppress TDP-43 aggregations

Apparently, there was no common rule for the effective combination between different Hero proteins and different client proteins. However, we noticed that Hero45, -7, and -11, which were particularly effective in preventing the cellular aggregation of TDP-43ΔNLS (Fig 4B), have unusually high positive charges (pI = 8.66, 10.44, and 11.55, respectively), with high

proportions of basic amino acid residues (proportion of Arg + Lys = 18.4%, 32.2%, and 27.3%, respectively). This raised a possibility that these positively charged properties may be important for their functions to prevent TDP-43ΔNLS aggregation. To confirm this idea, we constructed mutants of Hero7 and Hero11, in which Lys and Arg (positively charged) were replaced to Gly (non-charged). As shown in Fig 4C, lysine and arginine (KR)-less mutants of Hero7 and Hero11 have largely lost the anti-aggregation activity, suggesting that positive charges of Hero proteins are necessary for this activity, at least for TDP-43 aggregation.

To test the sufficiency of positive charges of Hero proteins, we next randomly scrambled the amino acid sequences of Hero7 and -11, and GST as a control, while keeping their original amino acid composition ratio (i.e., percentage of each amino acid) and fixing the total amino acid lengths at 100 amino acids (aa) or 42-aa (Fig 4D and S3 Table; three different shuffles for 100-aa and two different shuffles for 42-aa). Strikingly, the randomized 100-aa sequences, derived from extremely positively charged Hero7 and -11, prevented the aggregation of TDP-43ΔNLS with comparable or even higher efficiencies than the original ones. Importantly, none of the scrambled 42-aa sequences were able to prevent the TDP-43ΔNLS aggregation, regardless of their amino acid compositions. These results indicate that their amino acid composition and length (i.e., their molecular nature as long, hydrophilic, and highly charged "polymers"), but not their amino acid sequence per se, are the key for Hero proteins to impart their client-protecting functions, at least for TDP-43ΔNLS. Of note, it was previously shown that short peptide fragments of randomly chosen IDPs in humans (35–45 amino acids) have weak cryoprotective and lyophilization-protective activities (up to 1.3–1.5-fold compared to BSA) [26], suggesting that the amino acid length is also important for protecting other proteins under stress conditions.

## Hero proteins suppress the neurotoxicity caused by protein aggregates in motor neurons

We next examined the anti-aggregation activity of Hero proteins in human induced-pluripotent stem (iPS)-derived motor neurons. We expressed GFP-tagged TDP-43ΔNLS together with each of the six representative human Hero proteins or the GST negative control, as in HEK293T cells. Aggregates of GFP-tagged TDP-43ΔNLS gave saturated fluorescent signals, which were suppressed by co-expression of Hero7 (Fig 5A). To quantify the degree of aggregation, we defined the ratio of the saturated area as the aggregation index (Fig 5B). Similarly to the analysis in HEK293T cells (Fig 4B), Hero9, -45, -7, and -11 strongly reduced the aggregation formation in motor neurons (Fig 5C).

The neurite length is widely used as a measurement of the neuronal condition. Indeed, it is known that ALS patient–derived motor neurons with mutations in TDP-43 have shorter neurites compared to those from healthy donors [27]. To evaluate neuronal protective activity of Hero proteins, we measured the total neurite length of the transfected neurons. Except for Hero45, overexpression of Hero proteins themselves had no or little effect on neuronal morphology (Fig 5D). In contrast, overexpression of GFP-TDP-43ΔNLS markedly reduced the neurite outgrowth to approximately 30% compared to that of control neurons without aggregation (Fig 5F). However, co-expression of Hero9, -7, and -11 almost fully rescued the neurite outgrowth defects by TDP-43 aggregation (Fig 5E and 5F). Therefore, Hero proteins can efficiently protect human motor neurons from the neurotoxicity of TDP43 aggregation.

## Hero proteins suppress the neurotoxicity of protein aggregates in *Drosophila* eyes

Hero proteins' protective activity can be observed not only in cultured cells but also in *Drosophila* in vivo. The fly eye has been used as a valuable model system for studying neurodegenerative

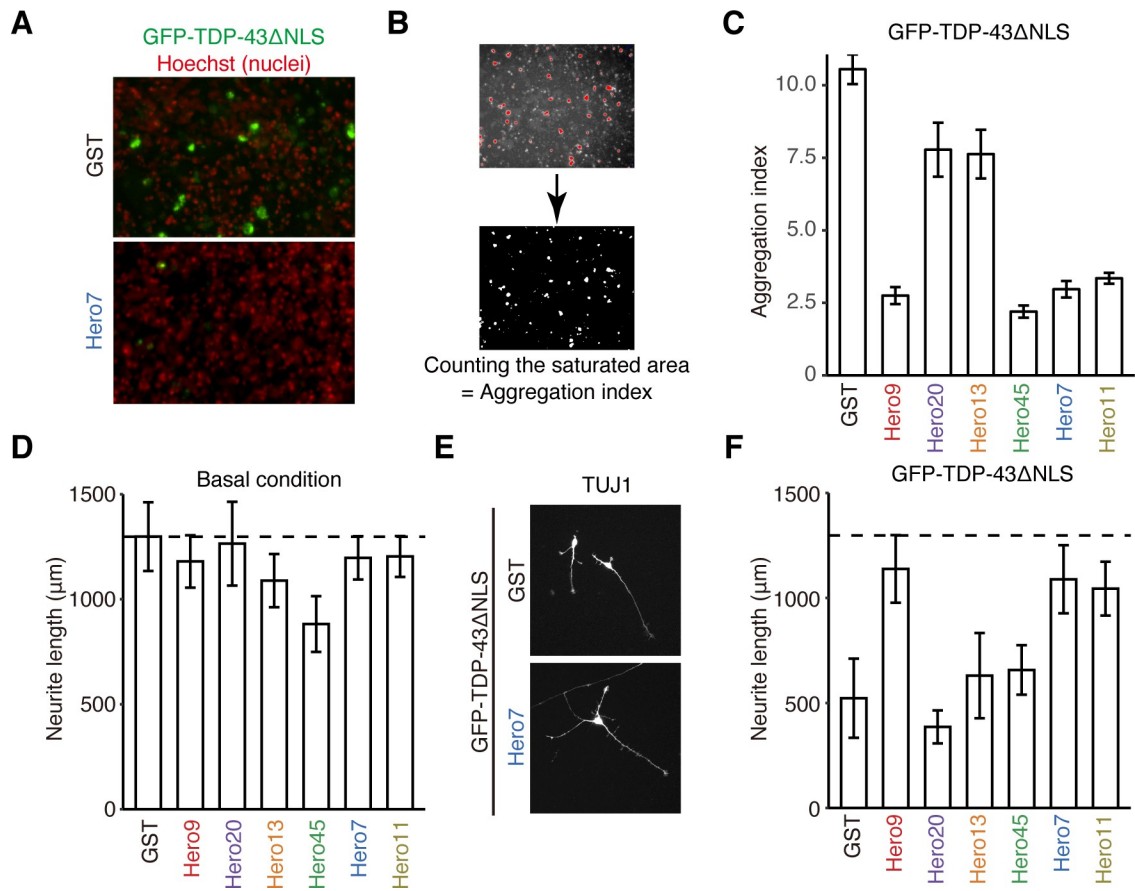

**Fig 5. Hero proteins suppress the neurotoxicity of TDP-43ΔNLS aggregates in motor neurons.** (A) Representative images of aggregation in human iPS-derived motor neurons transfected with GFP-TDP-43ΔNLS, together with the GST negative control or Hero7. Note that signals from aggregations are saturated. (B) Scheme for counting the saturated area of TDP-43ΔNLS aggregates. (C) Aggregation index analysis in motor neurons. GFP-TDP-43ΔNLS was expressed in motor neurons, together with each Hero protein or the GST negative control. Data indicate normalized saturated area (aggregation index), and represent means ± SD from 3 independent experiments. Hero9, -45, -7, and -11 reduced the aggregate formation. (D) Neurite length analysis in motor neurons. Each Hero protein or the GST control was individually expressed in motor neurons. Data indicate neurite length, and represent means ± SD from 8 cells. Hero45 showed a modest defect in neurite outgrowth. (E) Representative images of the neuron marker TUJ1 in motor neurons transfected with GFP-TDP-43ΔNLS, together with Hero7 or the GST control. Note that signals are saturated to clearly visualize the neurites. (F) Neurite length quantification in motor neurons transfected with GFP-TDP-43ΔNLS together with each Hero protein or the GST negative control. Data indicate neurite length, and represent means ± SD from 8 cells. Hero9, -7 and -11 rescued the neurite outgrowth defect by GFP-TDP-43ΔNLS aggregation. The numerical data pertaining to this figure can be found in S1 Data file. GFP, green fluorescence protein; GST, glutathione S-transferase; Hero, heat-resistant obscure; iPS, induced-pluripotent stem; TDP-43ΔNLS, TDP-43 lacking the nuclear localization signal; TUJ1, neuron-specific class III beta-tublin.

diseases. Indeed, expression of human TDP-43 fused with yellow fluorescent protein (YFP) (TDP-43-YFP) in the differentiating photoreceptor cells using a glass multimer reporter (GMR)-Gal4 driver causes retinal degeneration [28] (Fig 6A, middle). We generated transgenic flies expressing human Hero9, -13, -45 or -11 in the retina and introduced the transgenes into strains expressing TDP-43-YFP. Strikingly, the eye degeneration by TDP-43-YFP was almost completely suppressed by co-expression of Hero9 (Fig 6A, middle). The rescue effect of TDP-43-YFP by Hero45 was moderate but could be boosted by doubling the copy number of the transgene (Fig 6A, bottom). There were no harmful effects by retina-specific expression of Hero proteins per se; neither YFP nor any of the transgenic Hero proteins alone caused a detectable change in the normal eye morphology (Fig 6A, top). These data show that Hero proteins can suppress pathogenic protein aggregations in the living fly eye.

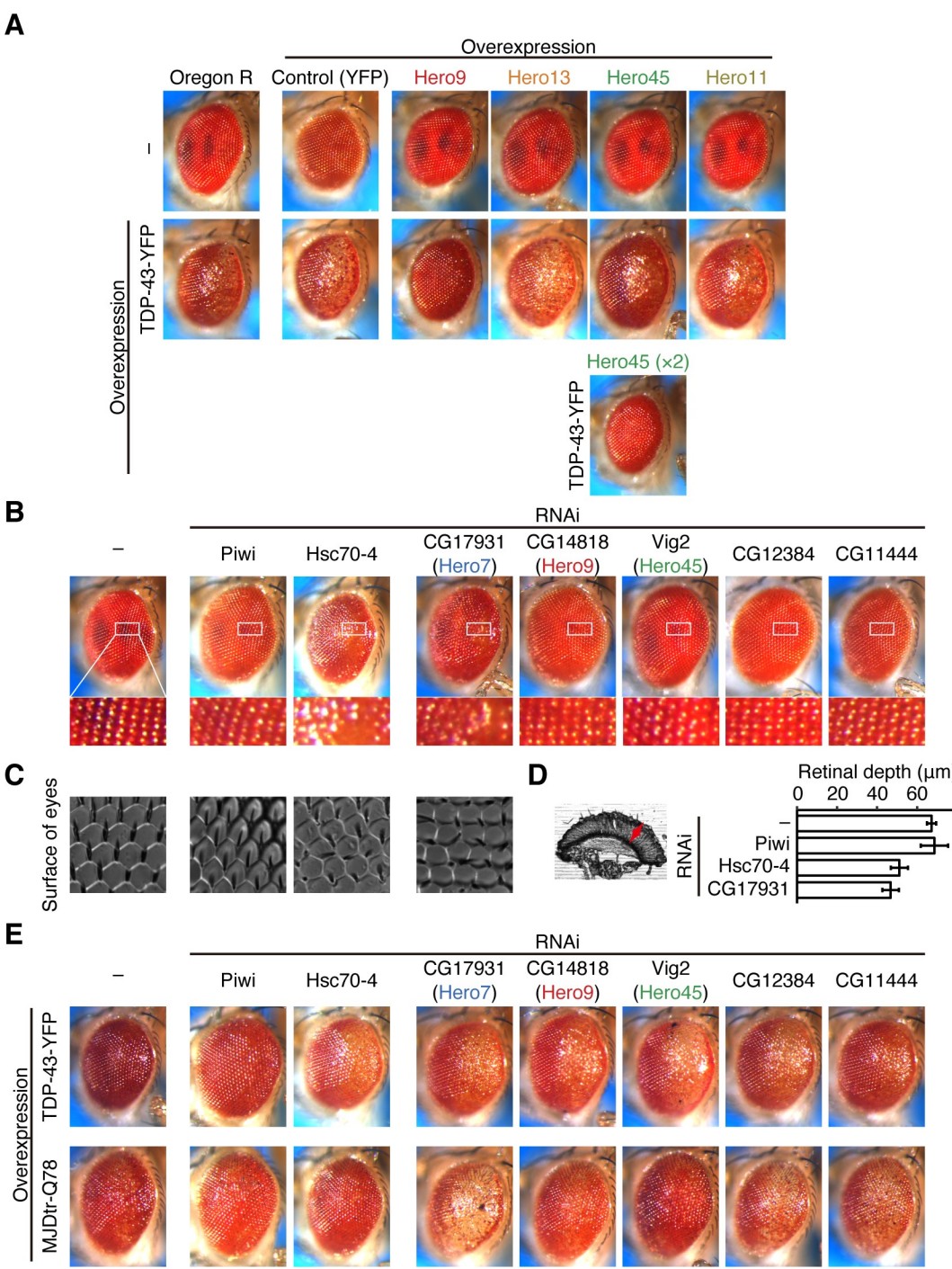

**Fig 6. Hero proteins suppress the neurotoxicity of protein aggregates in *Drosophila* eyes.** (A) Expression of Hero9, -13, -45, or -11 alone showed no phenotype in the external eye morphology, much as the YFP control (upper). Expression of TDP-43-YFP caused the external eye degeneration. Hero9 and double copies of Hero45 strongly suppressed the eye degeneration by TDP-43-YFP (middle and lower). (B) Knockdown of Hsc70-4 or CG17931 caused detectable external eye degeneration, compared to the mock knockdown of Piwi. (C) Nail polish imprint of eyes. Knockdown of Hsc70-4 or CG17931 caused irregular arrangements of ommatidia and bristles. (D) Knockdown of Hsc70-4 or CG17931 decreased the retinal depth. Arrowhead in the left picture indicates the retinal depth. Data represent means ± SD from 5 individual flies. (E) Knockdown Hsc70-4, CG17931, CG14818, or Vig2 caused exacerbation of the eye morphology defect by overexpression of TDP-43-YFP (upper). Knockdown of fly Hero proteins, CG17931, CG12384, or CG11444 caused exacerbation of the eye morphology defect by overexpression of aggregation-prone MJDtr-Q78 (lower). Note that female flies were used for (E), while male flies were used for (A–D). We observed the same phenotype for at least 10 flies per each genotype in each assay.

The numerical data pertaining to this figure can be found in S1 Data file. Hero, heat-resistant obscure; Hsc, heat shock cognate; MJDtr-Q78, Machado-Joseph disease truncated-poly-glutamine 78; RNAi, RNA interference; YFP, yellow fluorescent protein.

Using this *Drosophila* eye system, we next investigated functions of the endogenous Hero proteins. It is known that Hsc70-4, the most abundant and constitutively expressed Hsp70 chaperone, is required for normal eye development in *Drosophila* [29–31]. Indeed, knockdown of Hsc70-4 in the eyes showed a modest but detectable defect in external eye morphology, whereas mock depletion of germline-specific Piwi caused no abnormality (Fig 6B). Interestingly, eye-specific knockdown of CG17931, a homolog of human Hero7 (identity = 58%, similarity = 68%), also showed a similar eye degeneration phenotype as that of Hsc70-4. Moreover, irregular arrangements of ommatidia and bristles (Fig 6C), as well as reduction of the retinal depth (Fig 6D), were also caused by knockdown of Hsc70-4 or CG17931. These observations suggest that not only chaperones but Hero proteins also play important roles in fly eye development.

We then evaluated the protective effect of endogenous fly Hero proteins against pathogenic protein aggregates. To this end, we utilized the abovementioned aggregation-prone TDP-43-YFP (Fig 6E, upper) as well as human Ataxin3 containing a pathogenic expansion of 78 glutamine repeats (Machado-Joseph disease truncated-polyglutamine 78 [MJDtr-Q78]; Fig 6E, bottom), which also causes retinal degeneration when overexpressed in *Drosophila* eyes [32] (Fig 6E, bottom). Strikingly, defects by these aggregation-prone proteins were markedly exacerbated by simultaneous knockdown of the Hero7 homolog CG17931, compared to the negative control of Piwi knockdown (Fig 6E). Exacerbated eye phenotypes with MJDtr-Q78 and TDP-43-YFP were also apparent in the eye depleted of other endogenous fly Hero proteins, such as CG14818 and Vig2, which are potential homologs of human Hero9 (identity = 23%, similarity = 40%) and Hero45 (identity = 30%, similarity = 41%), respectively, and CG12384 and CG11444 (Fig 6E). Together, these data show that Hero proteins act to suppress aggregation-associated eye degeneration in living flies, consistent with our in cell data (Figs 4 and 5).

## Hero proteins are essential both in cells and in vivo

To understand Hero proteins' functions in non-overexpressed, physiological conditions, we performed clustered regularly interspaced short palindromic repeat (CRISPR)/Cas9-mediated knockout (KO) of six Hero proteins in HEK293T cells. We found that KO of Hero13 and 7 showed defects in cell proliferation under the normal culture condition (Fig 7A). Moreover, whole-body knockdown of endogenous Hero proteins in *Drosophila* using actin-GAL4-driven long hairpin RNAs often caused lethality (Fig 7B). Thus, Hero proteins play essential roles in human cell proliferation and *Drosophila* early development. We note that Frost in *Drosophila*, which is highly up-regulated after cold shock [33] and required for maintaining female fertility following cold exposure [34], is extremely disordered and charged, and presumably represents another example of Hero proteins with physiological functions in flies.

## Hero proteins can promote longevity in *Drosophila*

It is well known that aging is tightly associated to the loss of protein homeostasis [35,36]. For example, overexpression of Hsp70 significantly increases life expectancy in flies [37]. Such "longevity factors" are not limited to Hsp70 but include other chaperones and proteins related to autophagy and the Ub-proteasome system [35]. Strikingly, whole-body overexpression of endogenous Hero proteins in *Drosophila* often led to elongation of life span by more than

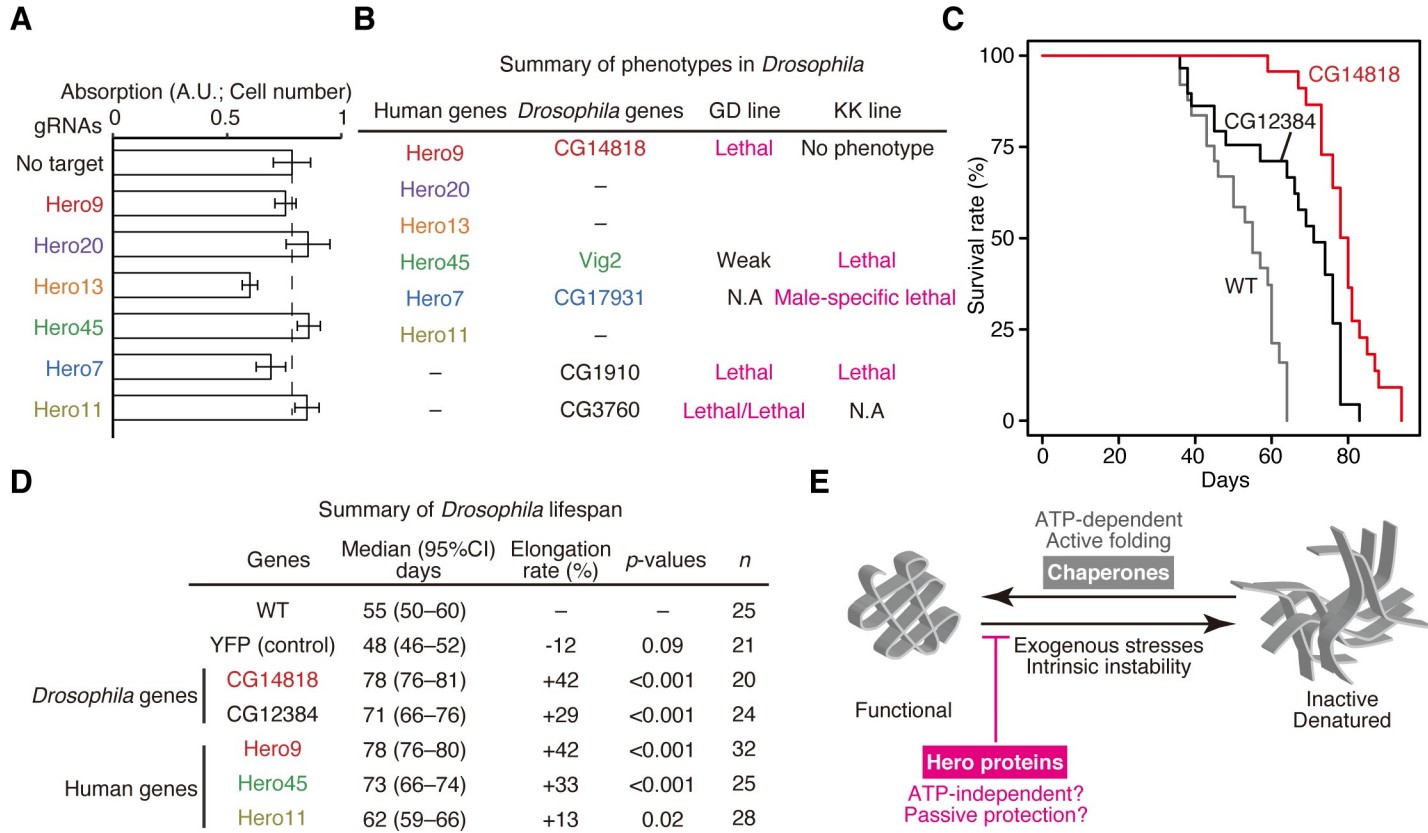

**Fig 7. Physiological functions of Hero proteins.** (A) Cell growth assay of HEK293T cells acutely knocked out for the indicated genes. Four days after transfection of gRNA, the numbers of the cells were measured by a coloring assay. Knockout of Hero13 and Hero7 reduced the cell proliferation. Data represent means ± SD of the cell numbers from 8 independent wells. (B) Whole-body knockdown phenotypes in *Drosophila*. Knockdown of endogenous fly Hero proteins were performed by two independent RNAi strains (GD and KK lines). Whole-body depletion of Hero proteins often caused lethality, suggesting that Hero proteins play essential roles in *Drosophila* early development. Note that we used two different GD lines for CG3760. We observed the same phenotype for at least two independent crosses per each genotype. (C) Survival plots of *Drosophila*. Whole-body overexpression of CG12384 or CG14818 elongated life spans, compared to the wild type. (D) Summary of life span analysis in *Drosophila*. Whole-body overexpression of Hero proteins except human Hero11 significantly elongated life spans. (E) A model for the modes of action of chaperones and Hero proteins. Chaperones usually act on the inactive/denatured state of their client proteins and actively reverse them to the functional state by using ATP, while Hero proteins likely act on the functional state of their client proteins and protect them from lapsing into the inactive/denatured state in an ATP-independent manner. The numerical data pertaining to this figure can be found in S1 Data file. gRNA, guide RNA; Hero, heat-resistant obscure; RNAi, RNA interference; WT, wild-type; YFP, yellow fluorescent protein.

approximately 30%, while YFP overexpression had no significant effect (Fig 7C and 7D). This was also true for interspecies overexpression of human Hero proteins in the fly whole body (Fig 7D). These observations support the idea that Hero proteins can combat against proteotoxic stress caused by aging.

## Discussion

Our study demonstrates that Hero proteins can stabilize various client proteins and protect them from denaturation or aggregation. Thus, not only plants, tardigrades, and other extremophiles, but non-extremophilic organisms such as humans universally utilize highly heat-resistant, hydrophilic, and charged proteins as molecular shields against protein instability. The simplest mode of action as molecular shields will be to reduce intermolecular collision/cohesion rates sterically and/or electrostatically, as previously proposed for LEA proteins [38]. However, this may not be the only mechanism for the client-protective activities. TDPs, LEA proteins, and Hero proteins belong to a large family of IDPs and can be regarded as their

extreme instances with extensive disorders. Currently, IDPs and proteins with IDRs are often linked to their abilities to cause liquid-liquid phase separation (LLPS), which is commonly thought to be a stage prior to aggregation. Indeed, we noticed that purified recombinant Hero7 protein spontaneously undergoes reversible gelation at a low temperature in vitro. Moreover, among different conditions tested in this study, there were some cases where Hero proteins promoted rather than inhibited aggregations, e.g., Hero20 for GA50 (Fig 4B). However, on the whole, Hero proteins tended to prevent aggregations of various proteins efficiently (Figs 3–7). This client-protective action of Hero proteins is seemingly opposite to the common notion of IDPs/IDRs as autonomous aggregation inducers. Notably, however, it was recently proposed that a prion-like IDR in yeast Sup35 acts to protect Sup35 itself against proteotoxic damage by extending its soluble/reversible phase space and thereby preventing it from irreversible aggregation [39,40]. Thus, it is tempting to speculate that Hero proteins can impart at least a part of their protective functions in a similar manner as the prion-like IDR in Sup35, except that they do so in trans on their client proteins.

*Trans*-acting protective functions of Hero proteins may be reminiscent of those of conventional molecular chaperones. Indeed, overexpression of Hero proteins protected the luciferase activity in HEK293T cells under heat shock, much as Hsp70 and Hsp90 chaperones (Fig 3D). Hero proteins also significantly increased the longevity of *Drosophila* (Fig 7C and 7D), as previously reported for Hsp70 [37]. Moreover, depletion of CG17931 (a homolog of human Hero7) and that of Hsc70-4 caused similar degeneration in the *Drosophila* eye (Fig 6B–6D). However, unlike molecular chaperones that often use cycles of ATP binding and hydrolysis, Hero proteins lack any apparent ATPase or any other domains. Moreover, chaperones usually act on the inactive/denatured state of their client proteins and actively reverse them to the functional state, whereas Hero proteins likely act on the functional state of their client proteins and protect them from lapsing into the inactive/denatured state (Fig 7E). We suggest that cooperation between chaperones and Hero proteins with distinct modes of action is central to the maintenance of the functional proteome. Moreover, it is also possible that some Hero proteins rely on classical chaperones for their activities or, conversely, some chaperones may be assisted by the protective effect of Hero proteins.

The fact that the primary sequences of Hero proteins are poorly conserved (Fig 2G and 2H) suggests that the molecular nature of Hero proteins as hydrophilic, charged, and disordered polymers is important for their functions. Supporting this idea, scrambling the amino acid sequences of Hero7 and Hero11 did not compromise their anti-aggregation activities for TDP-43ΔNLS in cells (Fig 4D). This observation challenges our conventional view of protein functions, in which the amino acid sequence of a protein determines its unique three-dimensional structure and thereby its function. Given the apparent importance of Hero proteins' physical properties, it is not surprising that other charged biopolymers such as nucleic acids are likely to play similar roles as protein stabilizers, as we observed (Fig 1B and 1E) and as reported recently [14,15].

Historically, the goal of biochemistry is to deconstruct biological events, identify enzymes responsible for the reactions of interest, and reconstitute them in a cell-free system. However, the enzymatic activity in reconstituted systems is often much less efficient than in crude cell lysate (Fig 1C and 1D), even when enzyme concentrations and buffer conditions are optimized. This may be at least in part explained by the absence of Hero proteins in classical reconstitution systems. If so, we might need to revise our approaches of biochemistry to take into consideration not only catalytic enzymes but also Hero proteins (and other environmental biomolecules). In addition, Hero proteins may be useful as stabilizers for protein pharmaceuticals, especially because those from humans are less likely to evoke immune responses than artificial polymers or biomolecules from other organisms. They may be co-expressed, used as additives

like polyethylene glycol (PEG), or can be fused to proteins of interest like PEGylation or PASy-lation (proline-alanine-serine) [41]. The finding that some Hero proteins can block pathogenic protein aggregates in cells and in fly models for neurodegenerative diseases may also serve as a starting point for future development of therapeutics or diagnostic methods.

Each organism appears to encode hundreds of Hero proteins with no apparent common motifs, and each Hero protein is likely to deal with many different client proteins, thus providing a rich but complex source to mine. Indeed, Hero45, which was originally identified as an mRNA-binding protein [42], has been found in the structure of the 80S ribosomes, where it occupies the mRNA-binding channel [43], but it is also known to be associated with hundreds of diverse proteins [44] with dynamic changes in its subcellular localizations in the cytoplasm, stress granules, nucleoli, etc. [45–47], highlighting the vast complexity of Hero proteins' actions. Moreover, individual Hero proteins show some preferences toward client proteins (Figs 3–6), and yet they may be at least partially overlapping in their functions. A combination of comprehensive approaches will be needed to decipher the global relationship among Hero proteins and their clients, including genetics to understand the physiological roles of Hero proteins, biochemistry and biophysics to dissect the molecular mechanisms of Hero protein actions, and evolutionary analyses to understand the birth and expansion of these genes.

## Materials and methods

### General methods

*Drosophila* S2 cells were cultured at 27˚C in Schneider's *Drosophila* Medium (Gibco) supplemented with 10% fetal bovine serum (Gibco) and Antibiotic-Antimycotic (Gibco). HEK293T cells were maintained in Dulbecco's Modified Eagle Medium (Sigma) supplemented with 10% FBS in 5% $CO_2$ atmosphere at 37˚C. Lysis buffer (30 mM HEPES-KOH pH 7.4, 100 mM KOAc, 2 mM Mg(OAc)$_2$), wash buffer (lysis buffer containing 1% Trition X-100 and 800 mM NaCl), and hypotonic lysis buffer (10 mM HEPES-KOH pH 7.4, 10 mM KOAc, 1.5 mM Mg (OAc)$_2$, 1× Complete EDTA-free protease inhibitor cocktail [Roche], and 5 mM DTT) were used. To prepare lysate from S2 cells or HEK293T cells, cells were collected and washed once with PBS. Cell pellets were resuspended with an equal volume of hypotonic lysis buffer, incubated for 15 minutes, vortexed for 15 seconds, and centrifuged at 17,000$g$ for 20 minutes. The supernatant was collected into tubes, shock-frozen in liquid nitrogen, and stored at −80˚C. For boiling, lysates were heated at 95˚C for 15 minutes, kept on ice for 1 minute, then centrifuged at 20,000$g$ for 5 minutes to remove aggregation.

### Plasmid construction

pCold-Hsc70-4, Hsp83, Hop, Droj2, p23, and pCAGEN-Flag-DEST were described previously [13].

### pCAGEN-FlagTevSnap-DEST and pCAGEN-GFP-DEST

A DNA fragment containing N-terminal FLAG-tag, TEV protease recognition sequence, and SNAP tag or the GFP sequence followed by the Gateway attR sites was inserted into pCAGEN by NEBuilder HiFi DNA Assembly Master Mix (NEB).

### pCAGEN-FlagTevSnap-Ago2(WT)

A DNA fragment containing a codon-optimized *Drosophila* Ago2 CDS sequence in pENTR/ D-Ago2 [11] was inserted into pCAGEN-*FlagTevSnap*-DEST using Gateway LR Clonase (Invitrogen).

### pCAGEN-Flag-GFP, pCAGEN-Flag-GST, and pCAGEN-FLuc

A DNA fragment containing GFP or GST with N-terminal FLAG-tag, or Firefly luciferase was inserted into pCAGEN by NEBuilder HiFi DNA Assembly Master Mix (NEB).

### pCAGEN-Flag-Hero9, -20, -13, -45, -7, -11, PGK1, IF4A1, GOT1, or Ub

A DNA fragment containing Hero9 (C9orf16), -20 (C11orf58), Hero13 (BEX3), Hero45 (SERBP1), Hero7 (SERF2), Hero11 (C19orf53), *PGK1*, *IF4A1*, *GOT1*, *or Ubiquitin* was amplified from HEK293T cell cDNA by PCR and cloned into pENTR/D-TOPO (Invitrogen), followed by recombination with pCAGEN-Flag-DEST using Gateway LR Clonase (Invitrogen).

### pCAGEN-HTTQ103-GFP

A DNA fragment containing HTT103Q-GFP was amplified from p426 103Q GPD (Addgene #1184; [21]) and inserted into pCAGEN by NEBuilder HiFi DNA Assembly Master Mix (NEB).

### pCAGEN-GFP-TDP43ΔNLS

A DNA fragment containing TDP-43 was amplified from HEK293T cell cDNA by PCR and cloned into pENTR/D-TOPO (Invitrogen), followed by recombination with pCAGEN-GFP-DEST using Gateway LR Clonase (Invitrogen). NLS deletion (78–99) in TDP-43 was performed by PCR-based site-directed mutagenesis.

### pCAGEN-GFP-GA50

A DNA fragment containing GA50 amplified from pAG303-Gal-GA50 (Addgene # 84907; [48]) and a DNA fragment containing GFP were inserted into pCAGEN by NEBuilder HiFi DNA Assembly Master Mix (NEB).

### pCold I-GFP, GST, Hero9, Hero20, -13, -45, -7, or -11-FlagHis

A DNA fragment containing CDS of each Hero protein and C-terminal FLAG- and His-tags was inserted into pCold I (Takara) by NEBuilder HiFi DNA Assembly Master Mix (NEB). For BEX3 and SERBP1, a codon-optimized DNA fragment was synthesized (Eurofins) and utilized as the template.

### pCAGEN-Flag-KR-less Hero7 or Hero11

A DNA fragment containing mutants of Hero7 or Hero11, in which positively charged amino acids (Lys and Arg) were substituted with neutral Gly, was inserted into pCAGEN-Flag-DEST vector.

### pCAGEN-Flag-shuffled 100[GST, Hero7, or Hero11]

A DNA fragment containing shuffled 100-aa sequences with the composition of GST, SERBP1, Hero7, or Hero11 was inserted into pCAGEN-Flag-DEST vector. The amino acid sequences are shown in S3 Table.

### pCAGEN-superFlag-shuffled 42[GST, Hero7, or Hero11]

A DNA fragment containing N-terminal superFlag [49] and shuffled 42-aa sequences with the composition of GST, SERBP1, Hero7, or Hero11 was inserted into pCAGEN vector. The amino acid sequences are shown in S3 Table.

## Protein purification

Recombinant proteins of Hsc70-4, Hsp83, Hop, Droj2, p23, and Dicer-2/R2D2 heterodimer were expressed and purified as described previously [13]. The same procedure was used to express and purify recombinant GFP proteins. Recombinant proteins of GST, Hero9, -20, -13, -45, -7, and -11 were expressed as C-terminal His-tagged proteins in *E. coli* Rosetta 2 (DE3) strain. Typically, the cells were cultivated in 1 L culture to an $OD_{600}$ of 0.4–0.6 at 37˚C with ampicillin and then grown at 15˚C overnight with 1 mM isopropyl-β-D-thiogalactoside (IPTG). The cell pellets were resuspend in His A buffer (30 mM HEPES-KOH pH 7.4, 200 mM KOAc, 2 mM Mg(OAc)$_2$, 5% glycerol) containing 1× EDTA-free protease inhibitor cocktail (Roche), sonicated, and centrifuged at 10,000*g* for 5 minutes twice. The supernatant was added to a slurry of cOmplete His-Tag Purification Resin (Roche) and incubated for 1 hour at room temperature or 2 hours at 4˚C, then eluted with His B buffer (His A buffer containing 400 mM imidazole). The eluents were collected and buffer exchanged to PBS with PD-10 (GE Healthcare).

## Ago2 elution assay

FLAG-TEV-SNAP-Ago2 expressed in HEK293T cells was labeled with SNAP-Surface Alexa Fluor 647 (NEB) and immunopurified by an anti-FLAG antibody conjugated onto Dynabeads Protein G (Invitrogen). The beads were washed three times by wash buffer, three times by wash buffer containing 2 mM ATP, and finally rinsed by lysis buffer. The beads were incubated with lysis buffer, crude lysates, or their boiled supernatants containing 2 U/μL TurboTEV protease (Accelagen) at room temperature for 1 hour. To prepare benzonase and proteinase K–treated (BZ/PK) supernatants, the supernatants were mixed with 1 U/mL benzonase (Millipore) and incubated at 37˚C for 1 hour, then mixed with 2 mg/mL proteinase K and incubated at 37˚C for 1 hour, and finally reboiled at 95˚C for 15 minutes to deactivate proteinase K. The benzonase or proteinase K treatment was omitted for the PK or BZ condition, respectively.

## Small RNA pull-down assay

Small RNA pull-down to detect pre-Ago2-RISC and mature Ago2-RISC formation was performed as described previously [13] with slight modifications. FLAG-TEV-Halo-Ago2 expressed in S2 cells was immunopurified by an anti-FLAG antibody conjugated onto Dynabeads Protein G (Invitrogen). The beads were washed three times by wash buffer, three times by wash buffer containing 2 mM ATP, and finally rinsed by lysis buffer. The Ago2-conjugated beads were then incubated with the reconstitution system, $^{32}$P-radiolabeled small RNA duplex, ATP regeneration system (1 mM ATP, 25 mM creatine monophosphate [Sigma], 0.03 U/μL creatine kinase [Calbiochem], and 0.1 U/μL RNasin Plus RNase Inhibitor [Promega]) and 3.5 μL of each boiled supernatant in 10 μL lysis buffer at 25˚C for 60 minutes. The reconstitution system contained 20 nM Dicer-2/R2D2, 600 nM Droj2, 1.8 μM Hsc70-4, 1.5 μM Hop, 3 μM Hsp83, and 3 μM p23. In every case, the total protein concentration was adjusted by the supplementation of GST. After incubation, beads were washed four times by wash buffer, then bound RNAs were extracted by proteinase K treatment and analyzed by native PAGE.

## Mass spectrometry

Proteins in the crude lysates and their boiled supernatants were trypsinized, desalted by ZipTip C18 (Millipore), concentrated, and subjected to nano liquid chromatography tandem mass spectrometry (nanoLC-MS/MS) analysis. For this analysis, we used LTQ-Orbitrap Velos mass spectrometer (Thermo Fisher Scientific) coupled with Dina-2A nanoflow LC system (KYA

Technologies). The samples were injected into 75-μm reversed-phase C18 columns at a flow rate of 10 μL/minute and eluted with a linear gradient of solvent A (2% acetonitrile and 0.1% formic acid in $H_2O$) to solvent B (40% acetonitrile and 0.1% formic acid in $H_2O$) at 300 nL/minute. The peptides were sequentially sprayed from nanoelectrospray ion source (KYA Technologies) and analyzed by the collision induced dissociation (CID) method. Mass spectra were acquired in data-dependent mode, switching automatically MS and MS/MS acquisition. All full-scan MS spectra in the range from $m/z$ 380 to 2,000 were acquired in the FT-MS part of the mass spectrometer with a target value of 1,000,000 and a resolution of 100,000 at $m/z$ 400. The 20 most intense ions that satisfied an ion selection threshold above 2,000 were fragmented in the linear ion trap with normalized collision energy of 35% for activation time of 10 ms. For accurate mass measurement, the Orbitrap analyzer was operated with the "lock mass" option using polydimethylcyclosiloxane ($m/z$ = 445.120025) and bis(2-ethylhexyl) phthalate ions ($m/z$ = 391.284286). The MS/MS signals were searched against the UniProt human proteome (*Homo sapiens*; UP000005640) and fly proteome (*Drosophila melanogaster*; UP000000803) using the Mascot algorithm (version 2.5.1; Matrix Science). Carbamidomethylation of cysteine was set as a fixed modification, whereas oxidation of methionine, protein N-terminal acetylation, and pyro-glutamination for N-terminal glutamine were set as variable modifications. Trypsin was defined as a proteolytic enzyme and a maximum of two missed cleavages were allowed. The mass tolerance was set to three parts per million (ppm) for peptide masses and 0.8 Da for MS/MS peaks. In the process of peptide identification, we conducted decoy database searches using Mascot and applied a filter to satisfy a false positive rate of less than 1%. One (1) was added to all the peptide spectrum match (PSM) values (to avoid zeros), which correlate with the protein amount, and the ratio between (PSM + 1) in the original crude lysate and that in the boiled supernatant was calculated in order to divide the identified proteins into 5 groups, each of which contained virtually the same number of proteins, according to the degree of enrichment or depletion by boiling.

## Prediction of structural disorders

Amino acid sequences were collected from UniProt human proteome (*H. sapiens*; UP000005640) and fly proteome (*D. melanogaster*; UP000000803). These sequences were subjected to the IUPred analysis [16]. The median IUPred score of all amino acid residues was allocated for each protein, and their distributions were plotted on histograms.

## Ortholog prediction

Ortholog prediction was performed by DIOPT, DRSC (*Drosophila* RNAi Screening Center) Integrative Ortholog Prediction Tool. The scores of best matched proteins between *Drosophila* and humans were plotted.

## Selection of representative Hero proteins

First, proteins with high IUPred scores (median of IUPred scores of all residues >0.6, and >75% of the residues have IUPred scores >0.4) and universally high expression profiles (median RPKM >25 in all tissues and cultured cells) were selected from the human protein atlas [18]. The proteins with clear organelle localization and/or known cellular functions were excluded. Finally, C9orf16 (Uniprot identifier; Q9BUW7), BEX3 (Q00994), C11orf58 (O00193), SERBP1 (Q8NC51), SERF2 (P84101), and C19orf53 (Q9UNZ5) were selected for further analyses.

## Western blot analysis

Anti-DDDDK antibody (M185, MBL) was used as the primary antibody at 1:10,000. Chemiluminescence was induced by Luminata Forte Western HRP Substrate (Millipore) and images were acquired by Amersham Imager 600 (GE Healthcare).

## Luciferase assay with a heat shock

HEK293T cells in a 96-well plate were transfected with 10 ng pCAGEN-FLuc and 90 ng empty pCAGEN or pCAGEN-Flag-Hero9, -20, -13, -45, -7, or -11 by using Lipofectamine 3000 (Thermo Fisher Scientific). After approximately 48 hours, 4 mM cycloheximide was added to the medium, and cells were incubated at 37°C for 10 minutes and then subjected to a heat shock at 45°C for 8 minutes. The cells were lysed by Passive Lysis Solution (Perkin Elmer) and incubated for 15 minutes at room temperature, and the lysates were collected. The luciferase activity of Fluc was measured by using sensilite Enhanced Flash Luminescence (Perkin Elmer).

## LDH dehydration assay

LDH from rabbit muscle (Roche) was diluted to 5 μg/mL in 5-fold diluted boiled supernatants (for Fig 1E), or PBS containing 4 μg/mL BSA, GST, or each Hero protein, 400 μg/mL Trehalose or 2 mg/mL Arginine (for Fig 2D). Samples were desiccated in SpeedVac vacuum concentrator (Thermo) for 16 hours without heating. For normalization, the same amount of LDH in the same conditions was kept on ice for the same duration as desiccation. Dried samples were rehydrated by PBS, and the LDH activity was measured by Cytotoxicity LDH Assay Kit-WST (Dojindo).

## Chloroform exposure assay of GFP

A total of 10 μg/mL GFP was mixed with 50 μg/mL BSA, GST, or each Hero protein, followed by the addition of the same volume of chloroform. The samples were continuously shaken at room temperate for 30 minutes and centrifuged at 10,000*g* for 1 minute. The GFP intensity in the supernatants (i.e., in the water phase) was measured.

## Filter trap assay of aggregates

HEK293T cells in a 12-well plate were transfected with 100 ng pCAGEN-GFP-HTTQ103, 100 ng pCAGEN-GFP-GA50, or 200 ng pCAGEN-GFP-TDP43ΔNLS together with 900 ng (for HTTQ103 and GA50) or 800 ng (for TDP43ΔNLS) of pCAGEN-Flag-Hero9, -20, -13, -45, -7, -11, GST, Luciferase, PGK1, IF4A1, GOT1, or Ubiqutin by using Lipofectamine 3000 (Thermo Fisher Scientific). After approximately 48 hours, the cells were resuspended in 200 μL PBS containing 1× Complete EDTA-free protease inhibitor cocktail (Roche) and sonicated by Bioruptor II (BMBio). The total protein concentrations were adjusted and then 1% SDS was added. Subsequently, a 5-fold volume of 1% SDS in PBS was added, then loaded onto a cellulose acetate membrane with a 0.2-μm pore size (GE Healthcare) that had been incubated in 1% SDS in PBS. After washes with 1% SDS, aggregates were detected by using anti-GFP antibody (Santa Cruz; sc-9996; 1:10,000).

## Microscopy

Cell images were acquired by an inverted type microscopy (IX83, Olympus) with a 20× objective lens (UCPLFLN 20×, 0.7 NA, Olympus).

## iPS-derived motor neuron differentiation and plasmid transfection

The procedure of motor neuron differentiation has been described previously [50]. Briefly, induced pluripotent stem cells (409B2 from Riken cell bank) were seeded to a Matigel-coated 12-well plate at a density of 70%–80% confluency in mTeSR plus medium (STEMCELL Technology) with 10 µM of Y-23632 (Rock inhibitor, Wako). After the cells reached to 95% confluent, the differentiation into neural lineage cells was initiated in DMEM/F12 (Sigma) supplemented with 15% KO serum replacement (KSR, Gibco), 1% GlutaMAX (Gibco), 1% nonessential amino acid (NEAA, Sigma), 10 µM of SB431542 (Wako), and 100 nM of LDN-193189 (Wako) for the initial 6 days of culture. For caudalization and ventralization, then, the cells were further supplemented with 1 µM of retinoic acid (RA) and 1 µM of smoothened agonist (SAG) and 10 µM of SU-5402 (Sigma), 10 µM of DAPT (Sigma), which accelerate neural differentiation [51] for 4 to 12 days of culture. SAG was synthesized with previously described protocol and purified by HPLC [52]. Then, the motor neurons (MNs) were maintained in Neurobasal medium (Thermo Fisher Scientific) supplemented with 2% B27 (Gibco), 1% GlutaMAX, 1% penicillin/streptomycin, and brain-derived neurotrophic factor (BDNF), maintenance medium. Before the Hero plasmid transfection, the MNs were re-seeded into a Matrigel-coated 96-well plate at a density of 2,000 cells/cm$^2$ dissociated by Accutase (Innovative cell technology) for 10–15 minutes. Then, after 5–7 days from reseeding, GFP-TDP-43ΔNLS (200 ng/wells) plasmid with and without each Hero plasmid (100 ng/wells) was transfected by Lipofectamine Stem Transfection Reagent in maintenance medium. Then, the aggregation, which is observed as a GFP-saturated area, was quantified after 48 hours from the transfection. To quantify the neurite length, the cells were fixed by 4% paraformaldehyde for 15 minutes and permeabilized by 0.1% Triton-X 100 for 5 minutes, followed by blocking with 1% BSA. The cells were then incubated with anti-TUJ1 mouse antibody (Biolegend, 1:1,000) in PBS at 4°C for overnight and then further incubated with Alexa Fluor 647–conjugated anti-mouse IgG (Thermo Fisher Scientific, 1:1,000) in PBS at 4°C for overnight with Hoechst 33342 for nuclear staining, following three times wash with PBS. The fluorescent images were acquired with a 20× objective by Axio observer (Zeiss).

## Image analysis for protein aggregation and quantifying neurite length

The images that show the exogenous expression of transfected GFP-TDP-43 were converted to 16-bit monochrome images and then masked according to saturated pixels (Fig 5B, red area). The images were binarized and calculated the total number of saturated GFP pixels. Aggregation index is defined by the percentage of the number of GFP-saturated pixels divided by total pixels in ROI. All post-image analysis was conducted by using Fiji (http://imagej.nih.gov/ij/). To quantify the neurite length and the number of branches of motor neuron, the neurites which are identified by TUJ1-positive features were traced by using the Simple Neurite Tracer plug-in (http://imagej.net/Simple_Neurite_Tracer) in Fiji.

## *Drosophila* eye degeneration assay

Fly culture and crosses were carried out at 25°C. All general fly stocks and RNAi lines were obtained from the Bloomington, Kyoto Fly Stock Center, FlyORF [53], or Vienna Drosophila Resource Center [54]. Flies carrying pUASg.attB-Hero9, -13, -45, -11, and Vig2 transgenes were generated by standard phiC31 integrase-mediated transgenesis (Bestgene). Gmr-Gal4 was used to express transgenes and/or induce RNAi by long hairpins in retina. UAS-MJDtr-Q78 [32], UAS-TDP-43-YFP, and UAS-YFP lines [28] were kindly provided by Dr. Nancy Bonini. Genotypes in Fig 6 were as follows: In (A), on the upper row, Oregon R is +/+;+/+. Control (YFP), Hero9, -13, -45, or -11 is *gmr-GAL4*; *UAS-YFP, UAS-Hero9, -13, -45*, or *-11/*

*SM6a-TM6B*. On the lower row,–is *gmr-GAL4*, *UAS-TDP-43-YFP*/+; *TM3*/+, Control (YFP) is *gmr-GAL4, UAS-TDP-43-YFP*/+; *UAS-YFP*/+, and Hero9, -13, -45, or -11 is *gmr-GAL4, UAS-TDP-43-YFP*/+; *UAS-Hero9, -13, -45, or -11*/+. Hero45(×2) is *gmr-GAL4, UAS-TDP-43-YFP*/+; *UAS-Hero45*/*UAS-Hero45*. External eyes were imaged in 1-day-old male adult flies. In (B),–is *gmr-GAL4*/+ and Piwi, Hsc70-4, CG17931, CG14818, or Vig2 is *gmr-GAL4/UAS-KK (Piwi), KK(Hsc70-4), KK(CG17931), KK(CG14818), or KK(Vig2)*. In (E),–is *gmr-GAL4, UAS-TDP43-YFP*/+ and Piwi, Hsc70-4, CG17931, CG14818, or Vig2 is *gmr-GAL4, UAS-TDP43-YFP /UAS-KK(Piwi), KK(Hsc70-4), KK(CG17931), KK(CG14818)*, or *KK(Vig2)* on the top row. In the bottom row,–is *gmr-GAL4*/+; *UAS-MJDtr-Q78*/+ and Piwi, Hsc70-4, CG17931, CG14818, or Vig2 is *gmr-GAL4/UAS-KK(Piwi), KK(Hsc70-4), KK(CG17931), KK(CG14818)*, or *KK(Vig2); UAS-MJDtr-Q78*/+. External eyes were imaged in 1-day-old female adult flies.

## Nail polish imprinting of *Drosophila* eyes

Imprints of *Drosophila* eyes by transparent nail polish were taken essentially as previously described [31,55], except that the imprints were mounted in 0.1% Tween 20. The preparations were examined using differential interference contrast (DIC) installed in DeltaVision Elite (GE healthcare).

## Imaging of *Drosophila* eye cryosections

Heads of wild-type and RNAi flies were fixed by 2.5% glutaraldehyde, 2% paraformaldehyde, and 0.1% Tween 20 in PBS at room temperature overnight. After washing with PBS, fixed fly heads were embedded in OCT compound (SAKURA Finetek, Japan) and subjected to cryosectioning. The internal retinal depth was quantified using ImageJ from cryosections at the same anatomical position for all animals. Five female heads were quantified from each genotype.

## Proliferation assay of acute-KO HEK293T cells

HEK293T cells stably expressing SpCas9 were established by a lentiviral system, using pLX_311-Cas9 (Addgene #96924) [56], psPAX2 (Addgene #12260), and pCMV-VSV-G (Addgene #8454) [57]. gRNAs were prepared as previously reported [58], and their sequences are listed in S4 Table. Cas9-expressing HEK293T cells in a 24-well plate were transfected with 50 ng of two different gRNAs (100 ng in total) for each target gene by using Lipofectamine RNAiMAX Transfection Reagent (Thermo Fisher Scientific). After approximately 72 hours, cell number was counted by using Cell Counting Kit-8 (Dojindo) according to the manufacturer's instruction.

## *Drosophila* lethality assay

Fly culture and crosses were carried out at 25˚C, and the viability was checked immediately after emergence. Actin-Gal4 was used to induce RNAi by long hairpins in the whole body. Genotypes in Fig 7B were as follows: *actin-GAL4/UAS-GD(CG14818), actin-GAL4/UAS-KK (CG14818), actin-GAL4/UAS-GD(CG11844; Vig2), actin-GAL4/UAS-KK(CG11844; Vig2), actin-GAL4/UAS-KK(CG17931), actin-GAL4*/+; *UAS-GD(CG1910)*/+, *actin-GAL4/UAS-KK (CG1910), actin-GAL4*/+; *UAS-GD(CG3760-1)*/+, *actin-GAL4*/+; *UAS-GD(CG3760-2)*/+.

## *Drosophila* life span assay

Male files were cultured at 25˚C. Every third or fourth day, dead flies were scored and the survivors were tipped into a fresh food vial. Kaplan-Meir survival curves were plotted and a log-rank test for trend was performed using R. Actin-Gal4 was used to express transgenes in the

whole body. Genotypes in Fig 7C were as follows: WT is *actin-Gal4/+*, YFP, Hero9, -45, -11, CG14818, or CG12384 is *actin-GAL4/UAS-YFP*, *Hero9*, *Hero45*, *Hero11*, *CG14818*, or *CG12384*.

## Supporting information

**S1 Fig. Prediction of disordered regions in Hero proteins.** (A) Amino acid sequences of the six representative Hero proteins. (B) Disorder prediction by IUPred using the default settings [16]. GFP and GST were used as controls. GFP, green fluorescent protein; GST, glutathione S-transferase; Hero, heat-resistant obscure.
(TIF)

**S2 Fig. Structured proteins have little effect on pathogenic protein aggregation in cells.** Filter trap assay of aggregation-prone proteins expressed in cells. TDP-43ΔNLS or HTTQ103 was expressed in a GFP-fusion form in HEK293T cells, together with each Hero protein, GST, or other structured proteins. The original or 5-fold diluted samples were loaded on a cellulose acetate membrane in the presence of 1% SDS, and the trapped aggregates were probed with anti-GFP antibody. We observed the same effects in two independent experiments. GFP, green fluorescent protein; GST, glutathione S-transferase; Hero, heat-resistant obscure; TDP-43ΔNLS, TDP-43 lacking the nuclear localization signal.
(TIF)

**S1 Table. Drosophila proteins identified by LC-MS/MS analysis of lysate and boiled supernatant.** LC-MS/MS, liquid chromatography-mass spectrometry.
(XLSX)

**S2 Table. Human proteins identified by LC-MS/MS analysis of lysates and boiled supernatant.** LC-MS/MS, liquid chromatography-mass spectrometry.
(XLSX)

**S3 Table. Shuffled amino acid sequences.**
(XLSX)

**S4 Table. gRNA sequences for acute KO.** gRNA, guide RNA; KO, knockout.
(XLSX)

**S1 Data. Excel spreadsheet containing the numerical values for each of the graphs represented in the manuscript.**
(XLSX)

**S1 Raw Images. This file contains the uncropped images of western blotting and filter trap assays in the manuscript.**
(PDF)

## Acknowledgments

We are grateful to Qinghua Liu for providing the plasmids for Dicer-2 and R2D2 expression; Connie Cepko for pCAGEN vector; Drosophila Genomics Resource Center, supported by NIH grant 2P40OD010949, for pUASg-HA-attB; Vienna Drosophila Resource Center for the RNAi lines; N. Bonini for UAS-SCA3trQ78, UAS-TDP43YFP, and UAS-YFP lines; Aaron Gitler for pAG303-Gal-GA50 (Addgene plasmid #84907); Susan Lindquist for p426 103Q GPD (Addgene plasmid #1184); Bob Weinberg for pCMV-VSV-G (Addgene plasmid #8454); Didier Trono for psPAX2 (Addgene plasmid #12260); John Doench, William Hahn, David

Root for pLX_311-Cas9 (Addgene plasmid #96924); and Yuko Fukuda for preparing cryosections. We thank Kaori Kiyokawa for experimental assistance, Hiro-oki Iwakawa for proposing the idea of boiling cell lysates, Life Science Editors for editorial assistance, and Phillip Zamore, Olivia Rissland, Shin-ichi Nakagawa, Hisashi Tadakuma, and the members of the Tomari laboratory for critical comments on the manuscript.

## Author Contributions

**Conceptualization:** Kotaro Tsuboyama, Eriko Matsuura-Suzuki, Shintaro Iwasaki, Yukihide Tomari.

**Data curation:** Kotaro Tsuboyama.

**Formal analysis:** Kotaro Tsuboyama, Tatsuya Osaki, Hiroko Kozuka-Hata, Masaaki Oyama, Yoshiho Ikeuchi.

**Funding acquisition:** Kotaro Tsuboyama, Yukihide Tomari.

**Investigation:** Kotaro Tsuboyama, Tatsuya Osaki, Hiroko Kozuka-Hata, Yuki Okada, Masaaki Oyama, Yoshiho Ikeuchi, Shintaro Iwasaki, Yukihide Tomari.

**Methodology:** Kotaro Tsuboyama, Tatsuya Osaki, Hiroko Kozuka-Hata, Masaaki Oyama, Yoshiho Ikeuchi.

**Project administration:** Yukihide Tomari.

**Resources:** Eriko Matsuura-Suzuki, Yuki Okada, Masaaki Oyama, Yoshiho Ikeuchi, Yukihide Tomari.

**Software:** Kotaro Tsuboyama.

**Supervision:** Eriko Matsuura-Suzuki, Yuki Okada, Masaaki Oyama, Yoshiho Ikeuchi, Shintaro Iwasaki, Yukihide Tomari.

**Validation:** Kotaro Tsuboyama.

**Visualization:** Kotaro Tsuboyama.

**Writing – original draft:** Kotaro Tsuboyama, Tatsuya Osaki, Hiroko Kozuka-Hata, Yuki Okada, Masaaki Oyama, Yoshiho Ikeuchi, Yukihide Tomari.

**Writing – review & editing:** Kotaro Tsuboyama, Shintaro Iwasaki, Yukihide Tomari.

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
