## [Editor Report · Decision Letter 0]

6 Nov 2019

Dear Yuki, 

Thank you for submitting your manuscript entitled "A widespread family of heat-resistant obscure (Hero) proteins protect against protein instability and aggregation" for consideration as a Research Article by PLOS Biology.

Your manuscript has now been evaluated by the PLOS Biology editorial staff as well as by an academic editor with relevant expertise. Althoug we and the academic editor find the conclusions very interesting and we were hoping that the academic editor would be able to make the call by looking at the rebuttal that you provided, s/he feels uncomfortable and would prefer to ask a couple of experts to decide. Thus we would like to send your submission out for external peer review and we will ask the reviewers to check the rebuttal as it would be a revision. I hope this is ok with you and, of course, we will try our best to conclude the process soon.

Before we can send your manuscript to reviewers, we need you to complete your submission by providing the metadata that is required for full assessment. To this end, please login to Editorial Manager where you will find the paper in the 'Submissions Needing Revisions' folder on your homepage. Please click 'Revise Submission' from the Action Links and complete all additional questions in the submission questionnaire.

Please re-submit your manuscript within two working days, i.e. by Nov 08 2019 11:59PM.

Kind regards,

Ines

--

Ines Alvarez-Garcia, PhD

Senior Editor

PLOS Biology

Carlyle House, Carlyle Road

Cambridge, CB4 3DN

+44 1223–442810

---

## [Decision Letter · Decision Letter 1]

29 Nov 2019

Dear Yuki,

Thank you very much for submitting your manuscript "A widespread family of heat-resistant obscure (Hero) proteins protect against protein instability and aggregation" for consideration as a Research Article at PLOS Biology. Your manuscript has been evaluated by the PLOS Biology editors, an Academic Editor with relevant expertise, and by two independent reviewers.

As you will see, the reviewers find your results novel and interesting, and recommend the manuscript for publication once you have addressed a few points. The reviewers ask for some clarifications, a request to make the data available to comply with our policies and one negative control that we deem essential. This control is needed to confirm that the observed effects do not reflect stochastic effects of the HERO proteins assayed. Please note that we won't require you to address the second request from Reviewer 1 to explore potential combinatorial effects of the HERO proteins, although we would welcome any data you migh have in hand.

In light of the reviews (attached below), we are pleased to offer you the opportunity to address the points from the reviewers in a revised version that we anticipate should not take you very long. We will then assess your revised manuscript and your response to the reviewers' comments.

Your revisions should address the specific points made by each reviewer. Please submit a file detailing your responses to the editorial requests and a point-by-point response to all of the reviewers' comments that indicates the changes you have made to the manuscript. In addition to a clean copy of the manuscript, please upload a 'track-changes' version of your manuscript that specifies the edits made. This should be uploaded as a "Related" file type. You should also cite any additional relevant literature that has been published since the original submission and mention any additional citations in your response. 

Before you revise your manuscript, please review the following PLOS policy and formatting requirements checklist PDF: http://journals.plos.org/plosbiology/s/file?id=9411/plos-biology-formatting-checklist.pdf. It is helpful if you format your revision according to our requirements - should your paper subsequently be accepted, this will save time at the acceptance stage.

IMPORTANT: Please note that as a condition of publication PLOS' data policy (http://journals.plos.org/plosbiology/s/data-availability) requires that you make available all data used to draw the conclusions arrived at in your manuscript. If you have not already done so, you must include any data used in your manuscript either in appropriate repositories, within the body of the manuscript, or as supporting information (N.B. this includes any numerical values that were used to generate graphs, histograms etc.). For an example see here: http://www.plosbiology.org/article/info%3Adoi%2F10.1371%2Fjournal.pbio.1001908#s5. Note that we will require a file with the data underlying the graphs shown in the following figures and make sure you indicate in each figure legend where the underlying data can be found:

Fig. 1D, E; Fig. 2E, F, G, H; Fig. 3B, C, D; Fig. 5C, D, F; Fig. 6D; Fig. 7A, C

For manuscripts submitted on or after 1st July 2019, we require the original, uncropped and minimally adjusted images supporting all blot and gel results reported in an article's figures or Supporting Information files. We will require these files before a manuscript can be accepted so please prepare them now, if you have not already uploaded them. Please carefully read our guidelines for how to prepare and upload this data: https://journals.plos.org/plosbiology/s/figures#loc-blot-and-gel-reporting-requirements.

Upon resubmission, the editors assess your revision and assuming the editors and Academic Editor feel that the revised manuscript remains appropriate for the journal, we may send the manuscript for re-review. We aim to consult the same Academic Editor and reviewers for revised manuscripts but may consult others if needed.

We expect to receive your revised manuscript within one month. Please email us (plosbiology@plos.org) to discuss this if you have any questions or concerns, or would like to request an extension. At this stage, your manuscript remains formally under active consideration at our journal; please notify us by email if you do not wish to submit a revision and instead wish to pursue publication elsewhere, so that we may end consideration of the manuscript at PLOS Biology.

When you are ready to submit a revised version of your manuscript, please go to https://www.editorialmanager.com/pbiology/ and log in as an Author. Click the link labelled 'Submissions Needing Revision' where you will find your submission record. 

Best wishes,

Ines

--

Ines Alvarez-Garcia, PhD

Senior Editor

PLOS Biology

Carlyle House, Carlyle Road

Cambridge, CB4 3DN

+44 1223–442810

Reviewers’ comments

Rev. 1:

The manuscript submitted to PLoS Biology by Tsuboyama et al. reports on an intriguing group of heat resistant, charged, and disordered proteins that exhibit protective effects on a variety of protein substrates. They also exhibit positive effects in cells and even organisms on aggregation and aging, respectively. The impetus for the study was the existence of protective proteins in radically distinct organisms and by virtue of the fact that a fusion protein containing Ago2 could be resolubilized after incubation with lysate. After isolation by resistance to severe heat denaturation from S2 cells and human (HEK293) tissue culture cells, the heat resistant lysate resolubilized Ago2 and reactivated the Ago2-containing RISC complex and rescued a model protein from dehydration. The isolated activity was partially protein and nucleic acid dependent. Mass spectroscopy was then used to isolate the surprising large number of proteins in the lysates (>900 in both S2 and HEK cells), which led to a bioinformatic analysis that helped characterize their features: lack of hydrophobicity, a high score in a composite disorder prediction, and the presence of positive and negative charges. Ultimately, the authors selected a group of “HERO” proteins exhibiting these properties. A subsequent analysis of the HERO proteins—individually—yielded fair to impressive effects (and even negative effects in some cases) in a range of assays/activities: protein aggregation, resistance to dehydration, and reactivation, as well as neuronal protein aggregation, neurite outgrowth, resistance to protein aggregation in the Drosophila eye, and Drosophila life span. Overall, an interesting collection of factors with a range of chaperone-like activities have been isolated through a careful analysis, one that encompasses a formidable amount of data.

Because the importance of this study relies not on the direct isolation of the HERO proteins but on the computational prediction of their existence, the study lacks a critical control: Instead of using GFP and GST as negative (arbitrary) controls, it was surprising that the investigators did not select proteins that were predicted, using their computational tools, to lack the desired properties inherent in the six selected HERO proteins. How might a collection of these negative controls behave in the assays employed? While this is not essential for every assay described in the paper, there are some experiments in which select HERO proteins are inactive, exhibit negative consequences, or at best exhibit ~2x improvements. How would a random collection of six proteins, or six proteins that lack the HERO attributes, behave in these assays? This is vital to buttress the central hypothesis of the study and to ensure that the observed effects do not reflect stochastic effects of the HERO proteins assayed.

Second, since the HERO proteins are evolutionarily diverse and are not related to previously identified LEA proteins, it is likely that they exhibit combinatorial effects. Have any pairs of the HEROs, or even the HERO collection, been examined? For the in vitro studies, this would be trivial, as subcritical concentrations—perhaps reflecting their cellular levels—could be combined with available preparations. For some of the cell-based studies, it would similarly be trivial to coexpress 2 or even 3 HERO proteins. Indeed, if the HERO proteins exhibit these protective functions in cells, they should work in combination to protect cellular proteostasis.

Other Comments:

1. In the cellular and organismal assays, the authors should discuss the potential that the HERO proteins rely on the activities of classical chaperones, many of which exhibit similar properties and endogenously expressed.

2. On p. 9, the authors propose that the disparate effects of the HERO proteins reflect distinct preferences for protein clients, but this does not explain why one HERO protein, SERF2/HERO7, exhibits pro-aggregation properties.

3. On p. 10, the authors refer to a previous study in which random IDPs were selected (Matsuo et al., 2018). But, what were the pI values for those IDPs examined in the study? Did they also exhibit a “bimodal distribution”?

Rev. 2: Edward Wallace – please note that this reviewer has waived anonymity

This manuscript identifies a class of protein chaperones that they call Heat-resistant obscure (Hero) proteins, that maintain their solubility in response to boiling, which chaperone the function of protein complexes (Ago2-RISC) without requiring ATP, and whose deletions give rise to phenotypes in drosophila and in human cell lines. The paper uses many different assays to carefully show the behaviour and importance of the Hero proteins. These observations are of major importance and wide interest, with potential applications as the authors discuss. It is also a cool story about how a “negative control” led to a new discovery.

The manuscript has already been through one round of review and the authors have responded thoroughly to those reviews, adding important and extensive new data. The work is clearly written, and of a high standard as well as of wide interest. I have three remaining concerns: first, the data availability is not acceptable for PLOS policies nor for community standards of FAIR science; second, biological replication is not clearly described; third, recent work on heat-denaturation of proteins at physiological temperatures needs to be addressed.

Data availability

The manuscript’s Data Availability Statement that "All data supporting the findings of this study are available from the corresponding authors on reasonable request,” does not comply with PLOS policy that "PLOS journals require authors to make all data underlying the findings described in their manuscript fully available without restriction, with rare exception.” There is no foreseeable reason for exception here. Currently, the supplementary data provided are inadequate.

I urge the authors to make their data available in their own interests, because well-organized open data increases the community’s trust in manuscript findings and thus increases citations, and is the best defence against the sadly common event where authors themselves might lose track of the data in future years. I urge them to read carefully PLOS policies on data availability, and also the excellent guide from the Leek lab: https://github.com/jtleek/datasharing.

Practically speaking, the key data are the proteomics LC-MS/MS data whose analysis is summarized in Figure 2. Raw data must be deposited in an archive such as PRIDE. The PSM data used to make Figure 2 must be included, in full, in a supplementary data file. This should be easy as the plots were clearly made in ggplot2 from data in tabular format, and indeed now it is easy to share both data and plotting code on platforms such as figshare. At a minimum, for all samples/fractions/replicates of both the drosophila and human datasets, the supplemental data must include (a) PSM counts for every detected protein (Figs 2A/B), (b) a table indicating the group for every detected protein (x axis, Figs 2C-J).

Methods must be revised so as to conform with the minimum information about a proteomics experiment guidelines (MIAPE; Taylor et al., Nat Biotechnol. 2007).

Replicates.

There is almost no mention of biological replicates or repeat experiments. Quantitative biochemical data, whether gels or proteomics, generally requires at least 2 independent biological replicates. The qualitative conclusions of the study are strong due to the many independent assays used that argue towards the same conclusions. For example, the proteomics data in two organisms and the gel/blot data presented overall agree that these obscure proteins are indeed heat-resistant. The recombinant FLAG-tag validation is clear including negative controls absent from the supernatant after boiling. The extensive functional validation of chaperone activity in vitro and in vivo is impressive. However the manuscript needs to state how many repeats there were, and acknowledge limitations in quantification as a result.

In the fly work, "Five female heads were quantified from each [eye cryosection] genotype.” How many flies were used in the other experiments, and how many replicate experiments?

Prior work on heat-denaturation (or not) of proteins

The abstract says "Proteins are typically denatured and aggregated by heat.” Then in introduction, “Proteins are generally vulnerable to heat, which destroys their structures and induces denaturation and aggregation.” Recent data counteract this long-established paradigm: importantly, Paola Picotti's lab published a careful proteome-wide stability analysis showing that many proteins are not denatured even far above physiological temperatures. Meanwhile, slight heat stress can cause specific protein aggregation that appears to be stress-protective rather than denaturation (reviewed in the citation by Franzmann and Alberti, as well as by Triandafillou et al., JBC 2019). Aggregation behaviour depends on both temperature and on factors such as pH and buffering conditions, as well as protein chaperones. The results statement "Many proteins are naturally unstable even under physiological conditions, probably because they need to prioritize function over structural durability,” could encompass this point that a protein's function could depend on its instability or aggregation.

The observations in this manuscript are new in that they reveal a class of proteins that maintain their solubility even at near-boiling temperatures very far above physiological temperature. Key points of the abstract and introduction should be moderated to reflect these recent findings and specify the range of temperatures involved.

Minor comments

In figure 2, why are some panels boxplots and others violin plots? These are two different ways of showing distributions, wouldn’t it be clearer to have all box or all violin?

It is confusing that TDP is used for both tardigrade disordered proteins and for TDP-43, can this be fixed?

Review response figures R1-R4 should be included as supplementary figures.

---

## [Editor Report · Decision Letter 2]

2 Jan 2020

Dear Yuki,

Thank you for submitting your revised Research Article entitled "A widespread family of heat-resistant obscure (Hero) proteins protect against protein instability and aggregation" for publication in PLOS Biology. I have now obtained advice from the Academic Editor and had the chance to discuss the revision with my colleagues. 

We're delighted to let you know that we're now editorially satisfied with your manuscript. However before we can formally accept your paper and consider it "in press", we also need to ensure that your article conforms to our guidelines. A member of our team will be in touch shortly with a set of requests. As we can't proceed until these requirements are met, your swift response will help prevent delays to publication. Please also make sure to address the data and other policy-related requests noted at the end of this email.

*Copyediting*

*Published Peer Review History*

*Early Version*

*Submitting Your Revision*

Best wishes,

Ines

--

Ines Alvarez-Garcia, PhD

Senior Editor

PLOS Biology

Carlyle House, Carlyle Road

Cambridge, CB4 3DN

+44 1223–442810

DATA POLICY:

Many thanks for including all raw data. Please add in Figure 7's legend where the underlying data can be found.

---

## [Editor Report · Decision Letter 3]

5 Feb 2020

Dear Dr Tomari,

On behalf of my colleagues and the Academic Editor, Raquel L. Lieberman, I am pleased to inform you that we will be delighted to publish your Research Article in PLOS Biology. 

Early Version

PRESS 

Kind regards,

Vita Usova 

Publication Assistant, 

PLOS Biology

on behalf of

Ines Alvarez-Garcia,

Senior Editor

PLOS Biology